# Trends in ambulance dispatches related to heat illness from 2010 to 2019: An ecological study

Daisuke Nakamura[1,2,3]*, Hiroki Kinoshita[1], Kazuo Asada[1], Takuma Arimitsu[4], Mikinobu Yasumatsu[2,3,5], Takayuki Ishiwata[2,5]

**1** Weathernews Inc., Makuhari Techno Garden, Chiba, Japan, **2** Rikkyo Research Institute of Wellness, Rikkyo University, Tokyo, Japan, **3** Physical Fitness Project, Japan Football Association (JFA), Tokyo, Japan, **4** Faculty of Health Care, Department of Human Health, Hachinohe Gakuin University, Aomori, Japan, **5** Department of Sport and Wellness, Rikkyo University, Saitama, Japan

* daisuke.nakamura@ac.cyberhome.ne.jp

**Data Availability Statement:** All ADRHI data are available from The Fire and Disaster Management Agency database (https://www.fdma.go.jp/disaster/heatstroke/post3.html). All WBGT data are available

## Abstract

Heatstroke is a serious heat-related illness that can even cause death. Heat alert systems play an important role in reducing the number of patients experiencing heat illness, as they encourage preventive actions such as the use of air conditioning, hydration, or other strategies. However, to date, the Japanese hazard classification has not considered seasonal and regional variations, despite clear differences in meteorological conditions across different regions in Japan. Moreover, several studies have reported a difference in thermoregulation between older and younger adults, implying that the hazard classification should also consider age differences. This study examined the relationship between the number of ambulance dispatches related to heat illness (ADRHI) and the Japanese heat hazard classification from 2010 to 2019, focusing on monthly and regional differences. Data from 47 prefectures during the 10-year period were collected and analyzed. ADRHI and wet bulb globe temperature (WBGT) data were collected from Japan's Ministry of Internal Affairs and Communications and the Ministry of the Environment Heat Illness Prevention Information website, respectively. The findings showed a significant relationship between ADRHI and $WBGT_{max}$ ($p < 0.05$, $r = 0.74$). ADRHI per 100,000 people showed significant differences across months. The post hoc test detected the first steep increase in ADRHI at a $WBGT_{max}$ of 23°C than at 22°C in June, and at a $WBGT_{max}$ of 26°C, 27°C, and 25°C in July, August, and September, respectively. Moreover, the first significant increase in ADRHI per 100,000 people at $WBGT_{max}$ differed across each region, at a $WBGT_{max}$ of 24°C in Hokkaido-Tohoku, 25°C in Kanto, Kansai, and Chugoku, 26°C in Chubu, 27°C in Shikoku, and 28°C in Kyushu-Okinawa. Further, Poisson regression analysis revealed that the relative risks differed across each region and month. These results imply that the hazard classification should be adjusted according to region and month in Japan.

from The Ministry of the Environment database (https://www.wbgt.env.go.jp/record_data.php?region=03&prefecture=44&point=44132). All population data are available from Ministry of Internal Affairs and Communications, Statistics Bureau (https://www.stat.go.jp/data/jinsui/2.html).

**Funding:** The author(s) received no specific funding for this work.

**Competing interests:** The authors have declared that no competing interests exist.

## Introduction

Heatstroke is known to be a serious heat-related illness that may even lead to death [1, 2]. Heat illnesses are complex conditions, and both intrinsic (e.g., health, hydration, acclimation) and extrinsic (e.g., environment, equipment, work-to-rest ratios) factors can influence risk [3]. Among these risk factors, there is a well-established association between high ambient temperature (i.e., heat) and higher rates of mortality and morbidity worldwide [4–7].

Although many studies have reported the relationship between the meteorological condition and the mortality related to heat illness [7–9], few studies have focused on the relationship between morbidity and heat illness [6, 10]. Ambulance dispatches related to heat illness (ADRHI) can approximately represent the morbidity of this condition. In fact, 692 Japanese people died of heat illnesses from June to September 2012 [11], whereas the total number of ADRHI was 45,701 [12], with a mortality rate of 1.5%. In addition, approximately 30% of all ADRHI in Japan's Kanto area was found in people older than 65 years (Old) [10].

Few studies have examined the relationship between ADRHI and the wet bulb globe temperature (WBGT). WBGT was designed to integrate the various meteorological variables that can influence human heat balance via gains or losses of energy through evaporative cooling (wet bulb temperature), radiant heating (globe temperature), and convection of sensible heat (dry bulb temperature). Currently, in Japan, WBGT is used as the index for heat alert systems.

Previous studies have reported the effectiveness of heat alert systems to decrease the number of heat illnesses. These systems are effective because they can both raise alertness about the possibility of heat illnesses and trigger preventive actions, such as the use of air conditioning, hydration, or other strategies [9, 13–15]. Thus, the criteria or cut-off values of these alert systems play an important role in heat illness prevention. For example, if the criteria do not match the actual mortality or morbidity related to heat illnesses, or when the actual meteorological conditions are different from the forecast, the effectiveness of these alert systems is likely to be weakened [9, 10]. However, despite the clear differences in meteorological condition throughout the year and among different regions in Japan, the hazard classification has not considered seasonal and regional differences. To support this issue, Ng et al. [10] further suggested that the hazard classification in Japan may not capture the vulnerabilities through which heat illnesses can occur at lower temperatures, which occur before the summer. The rainy season in Japan, which typically lasts from June to the end of July, is characterized by hot and humid environments, and people who are not acclimated to heat are exposed to a high risk of heat illnesses, even at a lower air temperature or WBGT. Aging contributes to a weakened heat-dissipation system, suggesting a higher risk for ADRHI in older rather than younger adults (YA) [16].

Given that the causal association between meteorological conditions and incidence of mortality has been demonstrated [7, 8, 17, 18], it is rational for governments and academic societies [19, 20] to have developed and implemented heat warning systems, such as heat alert systems and heat hazard levels, to prevent heat illnesses. For example, the US National Weather Service (NWS) issues heat advisories and warnings (collectively known as "heat alerts") in advance to forecast extreme heat events. Similarly, since 2006, the Ministry of the Environment of Japan has issued a hazard classification based on the WBGT with the aim to prevent heat illnesses, make people aware of heat events, and encourage preventive actions [21]. The hazard classification comprises five scales, ranging from "danger" (on days with WBGTs over 31°C) to "almost safe" (with WBGTs under 21°C). Moreover, since 2021, for days when the weather forecast indicates that the WBGT will be over <33°C, the ministry issues a heat alert either the day before or early in the morning on the relevant day. However,

this classification is not likely to match the actual number of ADRHI in Japan since it does not consider differences in meteorological conditions according to region or time of year.

Accordingly, exploring the characteristics of heat illness occurrences using the number of ADRHI could provide useful information to improve guidelines for hazard classification or heat alert systems. Moreover, although several studies have focused on the relationship between ADRHI and meteorological conditions in Japan, they did not use WBGT, even though this measurement is used in the hazard classification. Therefore, it is worth investigating the relationship between the occurrence of heat illness and the hazard classification—used for the prevention of heat illnesses in Japan—from the perspective of heat illness morbidity using big data analysis. This study aimed to examine the relationship between the number of ADRHI and the Japanese heat hazard classification from 2010 to 2019 in Japan, focusing on monthly and regional differences.

## Methods

### Study sites

To examine the relationship among the number of ADRHI and the WBGT data, we analyzed data from all prefectures and from the seven regions in Japan.

### Data sources

**Ambulance dispatches related to heat illness.** The data on ADRHI were acquired from the Fire and Disaster Management Agency under the Ministry of Internal Affairs and Communications, Japan [12]. They provided daily statistics of the number of people transported by ambulance in each of the 47 prefectures in Japan from June to September in every year from 2010 to 2019. As this period is summer in Japan, data on both ADRHI and WBGT are published. In this data set, the age of patients was recorded as follows: newborns are those aged less than 28 days, infants are aged from 28 days to 6 years, adolescents are aged from 7 to 17 years, adults are aged between 18 and 64 years, and older adults are aged 65 years and older. Days when no ADRHI transports were recorded were included in the analysis and counted as zero. Consequently, the total data set comprised 57,340 ADRHI and dispatches from 47 prefectures over 10 years.

**Wet bulb globe temperature.** The data on WBGT for the study period were acquired from the Ministry of Environment Heat Illness Prevention Information website [21]. Information on the exact time and location of each ADRHI in each prefecture was not available through the data set; thus, we used maximal WBGT data from the relevant meteorological station in the prefectural capital. As Tokyo was the only prefectural capital that did not have a meteorological station, we used data from the nearest station, which was about 10 km from the capital. In this study, the maximum WBGT of a given day was defined as $WBGT_{max}$. On the Ministry of the Environment Heat Illness Prevention Information website, four WBGT data points were not recorded; hence, we used 57,336 recordings of $WBGT_{max}$ in the analysis.

**Japanese heat hazard classification.** The Japanese heat hazard classification is defined by the following five categories: Danger, denoting that exercise is prohibited due to a WBGT $\geq$ 31˚C; Severe Warning, denoting that heavy exercise is prohibited due to a WBGT 28˚C–31˚C; Warning, denoting that rest should be provided often due to a WBGT 25˚C–28˚C; Caution, denoting that water should be replenished often due to a WBGT 21˚C–25˚C; and Almost Safe, denoting that appropriate water replenishment should be provided due to a WBGT < 21˚C. This heat hazard classification is published by the Japan Sports Association [22] and used by the Ministry of the Environment to prevent excessive heat illnesses.

**Population data and regional settings.** We acquired population data from each prefecture from the Ministry of Health, Labour and Welfare website [23], which uploads data annually. Table 1 shows the average population (in thousands) in all prefectures, the seven regional settings during the study period, and locational data.

## Statistical analysis

The WBGT was divided per 1°C, from 20°C to 35°C. The ADRHI was evaluated by absolute value and per 100,000 people (ADRHI_ALL), per 100,000 people aged 65 and older (ADR-HI_OLD), and other than ADRHI_OLD (ADRHI_YA). To analyze the relationship between ADRHI and WBGT, we examined Spearman's correlation coefficient by TIBCO Spotfire® [24]. To compare the differences for the number of ADRHI by month and region, we conducted the Kruskal–Wallis test. When a significant difference was detected, the Dann-Bonferroni test was used for post hoc analysis. In addition, to clarify the relationship between ADRHI and $WBGT_{max}$, the $WBGT_{max}$ of the first significant increase in ADRHI compared to the previous $WBGT_{max}$ was defined as the first significant increase point. Moreover, to investigate the effects on ADRHI of a one unit increase in WBGT in each region, a Poisson regression analysis was performed using SPSS (version 27.0; SPSS Inc, Chicago, IL). In the Poisson model, the ADRHI was set as the response value, and regions, month, WBGTmax, and proportion of people aged over 65 were set as a predictor. Population size was set as an offset value. The statistical significance was set at $p < 0.05$.

## Results

Table 2 shows the total annual number of ADRHI and the average total annual number of ADRHI_ALL throughout the study period in each prefecture. The highest total annual number of ADRHI per year appeared in Tokyo, with 4,286.7 cases, while the highest total annual number of ADRHI_ALL appeared in Okayama, having 66.0 cases throughout the study period (Table 2). Fig 1 shows the changes in ADRHI by year. The highest number of ADRHI appeared in 2018 (92,710 cases) and the lowest appeared in 2014 (40,048 cases). Fig 2 shows the ADRHI ratios stratified by age and year.

Fig 3 shows the relationship between the number of ADRHI and $WBGT_{max}$. A significant relationship between ADRHI and $WBGT_{max}$ ($p < 0.05$, r = 0.74, CI: 0.736–0.744, n = 57,336) was observed. The distribution of $WBGT_{max}$ stratified by month and region is shown in Figs 4 and 5, respectively. The $WBGT_{max}$ in each month differed significantly, with the differences in all months having differed significantly from other months ($p < 0.05$, Fig 4). The $WBGT_{max}$ also differed significantly between regions ($p < 0.05$, Fig 5).

Fig 6 shows ADRHI_ALL stratified by month. The values were 0.12 ± 0.20 (CI: 0.117–0.123), 0.67 ± 0.75 (CI: 0.659–0.683), 0.66 ± 0.64 (CI: 0.645–0.666), and 0.12 ± 0.23 (CI: 0.117–0.125) in June, July, August, and September, respectively. A significant difference was observed across the months, with the post hoc test detecting that all months differed significantly from other months ($p < 0.05$, Fig 6). The first steep increase in ADRHI_ALL was seen at a $WBGT_{max}$ of 23°C rather than at 22°C in June, and at a $WBGT_{max}$ of 26°C, 27°C, and 25°C in July, August, and September, respectively, when compared to the previous $WBGT_{max}$ (Fig 7).

No significant difference was observed between regions in the number of ADRHI_ALL throughout the study period ($p > 0.05$). Conversely, when the data were analyzed across regions, the first significant increase in ADRHI_ALL at a $WBGT_{max}$ differed across each region, at a $WBGT_{max}$ of 24°C in Hokkaido-Tohoku, 25°C in Kanto, Kansai, and Chugoku, 26°C in Chubu, 27°C in Shikoku, and 28°C in Kyushu-Okinawa, compared to the previous $WBGT_{max}$ (Table 3). The first steep increases in the number of ADRHI_ALL, ADRHI_OLD,

**Table 1. Average population (in thousands) data for all prefectures and seven regions.**

| Regions | Prefectures | Population (in thousands) | Longitude | Latitude |
|---|---|---|---|---|
| Hokkaido-Tohoku | Hokkaido | 5387 | 43.06 | 141.35 |
| Hokkaido-Tohoku | Aomori | 1313 | 40.82 | 140.74 |
| Hokkaido-Tohoku | Iwate | 1279 | 39.7 | 141.15 |
| Hokkaido-Tohoku | Miyagi | 2327 | 38.27 | 140.87 |
| Hokkaido-Tohoku | Akita | 1029 | 39.72 | 140.1 |
| Hokkaido-Tohoku | Yamagata | 1126 | 38.24 | 140.36 |
| Hokkaido-Tohoku | Fukushima | 1927 | 37.75 | 140.47 |
| Kanto | Ibaraki | 2918 | 36.34 | 140.45 |
| Kanto | Tochigi | 1974 | 36.57 | 139.88 |
| Kanto | Gunma | 1975 | 36.39 | 139.06 |
| Kanto | Saitama | 7262 | 35.86 | 139.65 |
| Kanto | Chiba | 6223 | 35.6 | 140.12 |
| Kanto | Tokyo | 13488 | 35.69 | 139.69 |
| Kanto | Kanagawa | 9115 | 35.45 | 139.64 |
| Chubu | Niigata | 2305 | 37.9 | 139.02 |
| Chubu | Toyama | 1069 | 36.7 | 137.21 |
| Chubu | Ishikawa | 1155 | 36.59 | 136.63 |
| Chubu | Fukui | 788 | 36.07 | 136.22 |
| Chubu | Yamanashi | 838 | 35.66 | 138.57 |
| Chubu | Nagano | 2103 | 36.65 | 138.18 |
| Chubu | Gifu | 2035 | 35.39 | 136.72 |
| Chubu | Shizuoka | 3705 | 34.98 | 138.38 |
| Chubu | Aichi | 7476 | 35.18 | 136.91 |
| Chubu | Mie | 1820 | 34.73 | 136.51 |
| Kansai | Shiga | 1413 | 35 | 135.87 |
| Kansai | Kyoto | 2611 | 35.02 | 135.76 |
| Kansai | Osaka | 8838 | 34.69 | 135.52 |
| Kansai | Hyogo | 5535 | 34.69 | 135.18 |
| Kansai | Nara | 1368 | 34.69 | 135.83 |
| Kansai | Wakayama | 965 | 34.23 | 135.17 |
| Chugoku | Tottori | 574 | 35.5 | 134.24 |
| Chugoku | Shimane | 696 | 35.47 | 133.05 |
| Chugoku | Okayama | 1921 | 34.66 | 133.93 |
| Chugoku | Hiroshima | 2837 | 34.4 | 132.46 |
| Chugoku | Yamaguchi | 1406 | 34.19 | 131.47 |
| Shikoku | Tokushima | 759 | 34.07 | 134.56 |
| Shikoku | Kagawa | 978 | 34.34 | 134.04 |
| Shikoku | Ehime | 1389 | 33.84 | 132.77 |
| Shikoku | Kochi | 732 | 33.56 | 133.53 |
| Kyushu-Okinawa | Fukuoka | 5094 | 33.61 | 130.42 |
| Kyushu-Okinawa | Saga | 834 | 33.25 | 130.3 |
| Kyushu-Okinawa | Nagasaki | 1380 | 32.74 | 129.87 |
| Kyushu-Okinawa | Kumamoto | 1786 | 32.79 | 130.74 |
| Kyushu-Okinawa | Oita | 1168 | 33.24 | 131.61 |
| Kyushu-Okinawa | Miyazaki | 1107 | 31.91 | 131.42 |
| Kyushu-Okinawa | Kagoshima | 1657 | 31.56 | 130.56 |
| Kyushu-Okinawa | Okinawa | 1426 | 26.21 | 127.68 |

**Table 2. The average total number of ADRHI per year (ADRHI / year) and average total number of ADRHI per 100,000 people / year (ADRHI_ALL / year) throughout the 10-year (2010–2019) study period, stratified by prefecture.**

| Regions | Prefectures | ADRHI / year | ADRHI_ALL / year |
|---|---|---|---|
| Hokkaido-Tohoku | Hokkaido | 988.1 | 18.4 |
| Hokkaido-Tohoku | Aomori | 375.3 | 28.6 |
| Hokkaido-Tohoku | Iwate | 485.4 | 38.0 |
| Hokkaido-Tohoku | Miyagi | 916.9 | 39.4 |
| Hokkaido-Tohoku | Akita | 439.2 | 42.8 |
| Hokkaido-Tohoku | Yamagata | 455.1 | 40.6 |
| Hokkaido-Tohoku | Fukushima | 1023.2 | 53.3 |
| Kanto | Ibaraki | 1418.3 | 48.7 |
| Kanto | Tochigi | 899.6 | 45.6 |
| Kanto | Gunma | 1133.9 | 57.5 |
| Kanto | Saitama | 3539.0 | 48.7 |
| Kanto | Chiba | 2546.6 | 40.9 |
| Kanto | Tokyo | 4286.7 | 31.7 |
| Kanto | Kanagawa | 2720.3 | 29.8 |
| Chubu | Niigata | 1165.2 | 50.6 |
| Chubu | Toyama | 387.0 | 36.3 |
| Chubu | Ishikawa | 525.7 | 45.6 |
| Chubu | Fukui | 350.5 | 44.6 |
| Chubu | Yamanashi | 375.7 | 45.0 |
| Chubu | Nagano | 782.8 | 37.3 |
| Chubu | Gifu | 1059.0 | 52.2 |
| Chubu | Shizuoka | 1467.7 | 39.7 |
| Chubu | Aichi | 3689.1 | 49.3 |
| Chubu | Mie | 987.6 | 54.4 |
| Kansai | Shiga | 654.9 | 46.3 |
| Kansai | Kyoto | 1518.4 | 58.2 |
| Kansai | Osaka | 3850.1 | 43.6 |
| Kansai | Hyogo | 2601.5 | 47.1 |
| Kansai | Nara | 764.8 | 56.1 |
| Kansai | Wakayama | 602.7 | 62.6 |
| Chugoku | Tottori | 351.0 | 61.4 |
| Chugoku | Shimane | 378.1 | 54.4 |
| Chugoku | Okayama | 1266.6 | 66.0 |
| Chugoku | Hiroshima | 1405.9 | 49.6 |
| Chugoku | Yamaguchi | 615.1 | 43.9 |
| Shikoku | Tokushima | 382.0 | 50.6 |
| Shikoku | Kagawa | 565.3 | 57.9 |
| Shikoku | Ehime | 761.1 | 55.0 |
| Shikoku | Kochi | 457.0 | 62.7 |
| Kyushu-Okinawa | Fukuoka | 2175.5 | 42.7 |
| Kyushu-Okinawa | Saga | 484.5 | 58.3 |
| Kyushu-Okinawa | Nagasaki | 695.0 | 50.6 |
| Kyushu-Okinawa | Kumamoto | 1104.4 | 62.0 |
| Kyushu-Okinawa | Oita | 627.2 | 53.9 |
| Kyushu-Okinawa | Miyazaki | 607.9 | 55.2 |
| Kyushu-Okinawa | Kagoshima | 1039.7 | 63.1 |
| Kyushu-Okinawa | Okinawa | 753.3 | 52.7 |

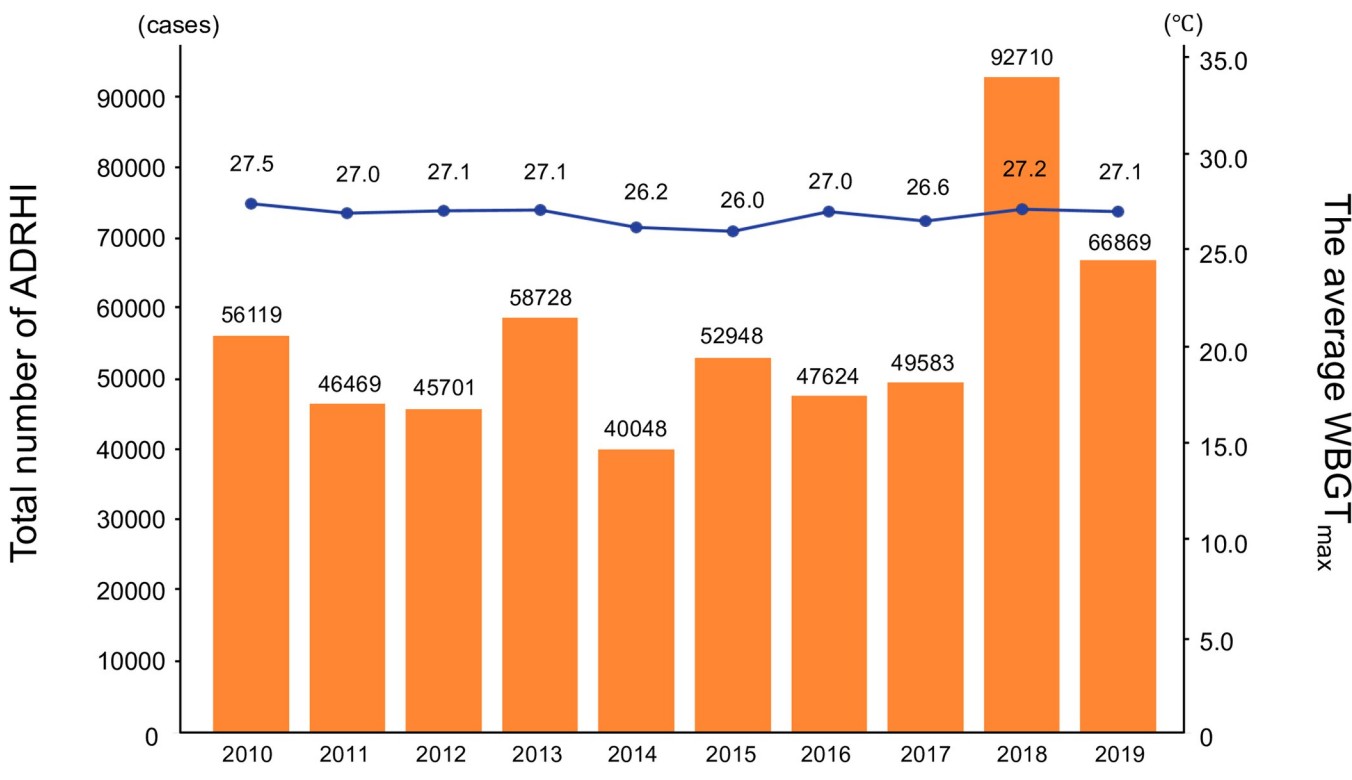

**Fig 1. Changes in ADRHI and average WBGT$_{max}$ in each year throughout the study period.**

and ADRHI_YA at WBGT$_{max}$ of all regions are shown in Table 3. Moreover, the relative risks differed for each region and month are shown in Table 4 (LR$\chi$2 = 697742.8, p < 0.001). The relative risks also differed among regions and months. Fig 8 shows the ratio of ADRHI in each hazard level in each region and month. In June, approximately 64.1% of the ADRHI that occurred in the Hokkaido-Tohoku region were related to a WBGT$_{max}$ of > 25˚C, which is categorized as "Caution"—the 2nd lowest level in the hazard classification, whereas only 15.0% of ARDHI occurred in the "Caution" level at Kyushu-Okinawa. However, from July to September, most ADRHI occurred at levels over "Warning".

## Discussion

This study aimed to examine the relationship between the number of ADRHI and WBGT data from 2010 to 2019 in Japan, focusing on monthly and regional differences.

To the best of our knowledge, this was the first study to examine the relationship between ADRHI and WBGT in Japan using big data analysis. As expected, the number of ADRHI showed a significant relationship with WBGT$_{max}$ (p < 0.05, r$^2$ = 0.55; Fig 3). This result is similar to the previous study, which reported that r$^2$ was 0.48 in three big prefectures in Japan [6]. During our experimental period, the average WBGT$_{max}$ was highest in July 2018, while the number of ADRHI cases were highest in 2018.

Moreover, our findings showed that among all ADRHI, YAs accounted for 50% of the heat illness (Fig 2); this value was much bigger than the average of 30% in big cities in Kanto shown in a prior study [10]. This difference between studies may be because of different data sources and study settings; the current study collected the ADRHI data from the Fire and Disaster Management Agency in Japan, whereas the cited previous study collected these data from fire

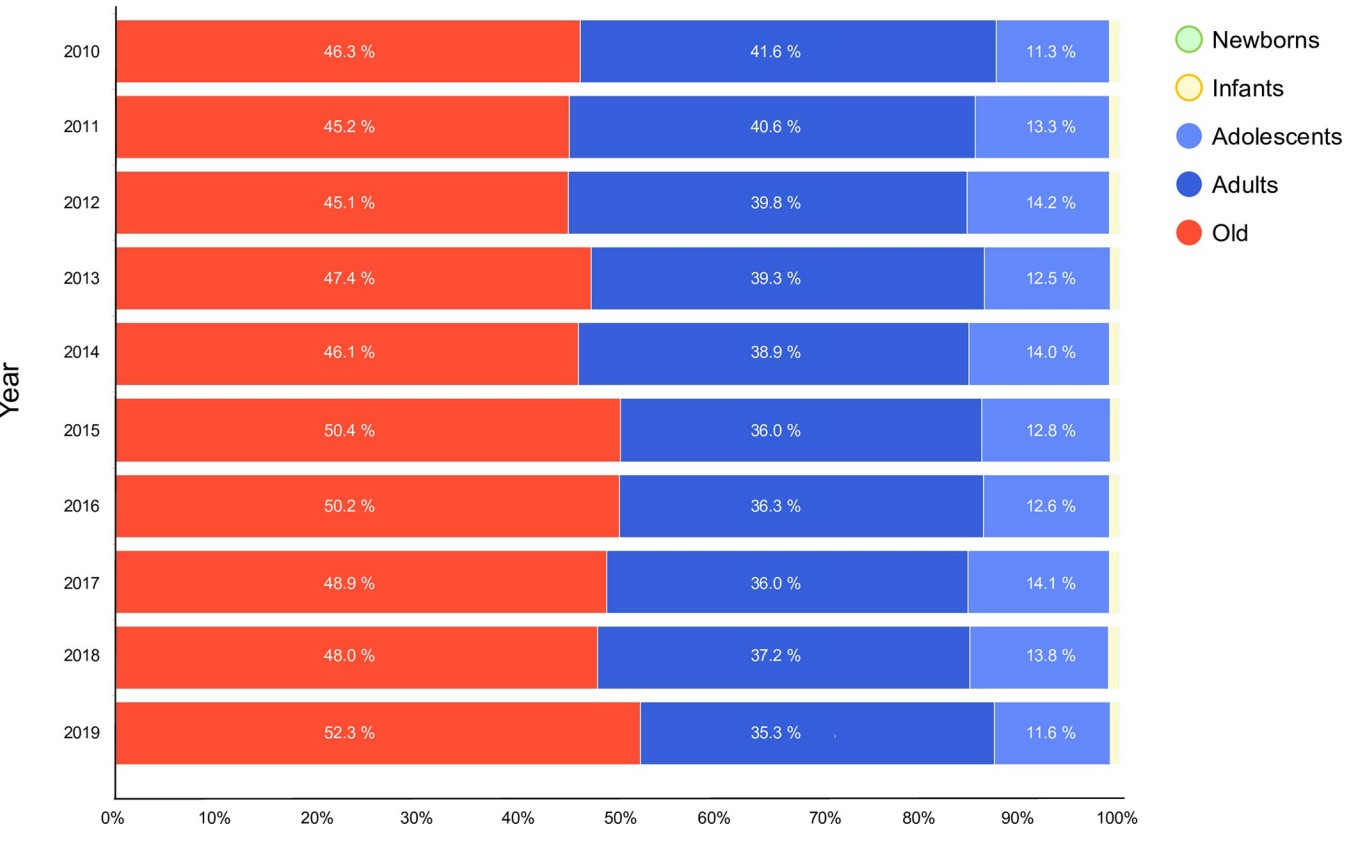

**Fig 2. ADRHI ratios stratified by age and year.**

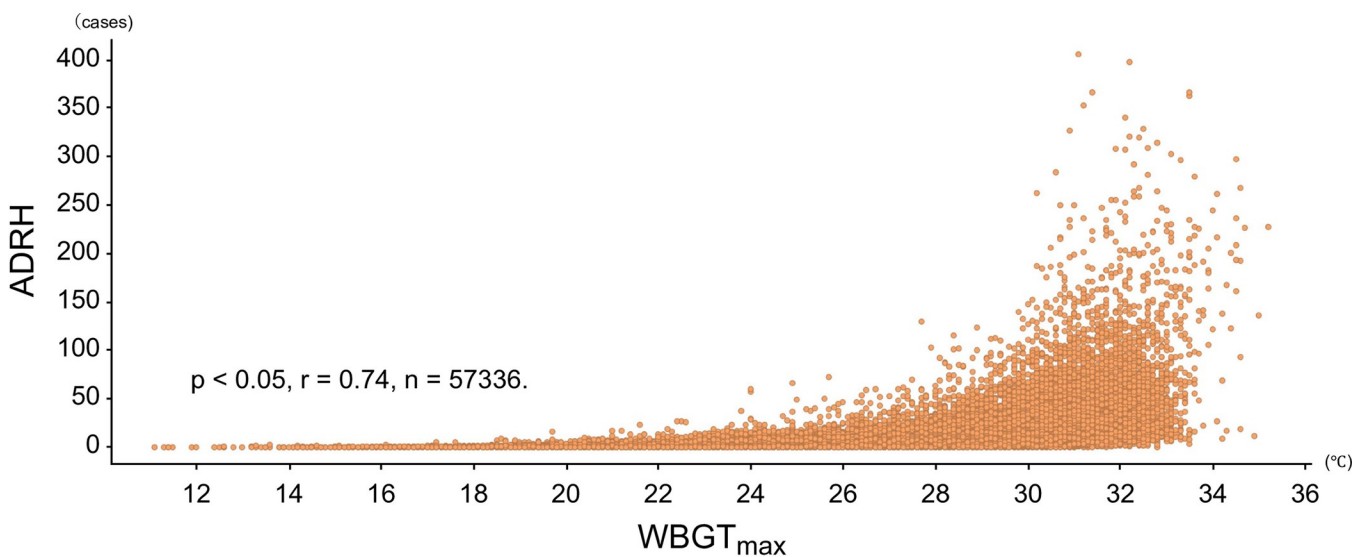

**Fig 3. The relationship between the number of ADRHI and WBGT$_{max}$.**

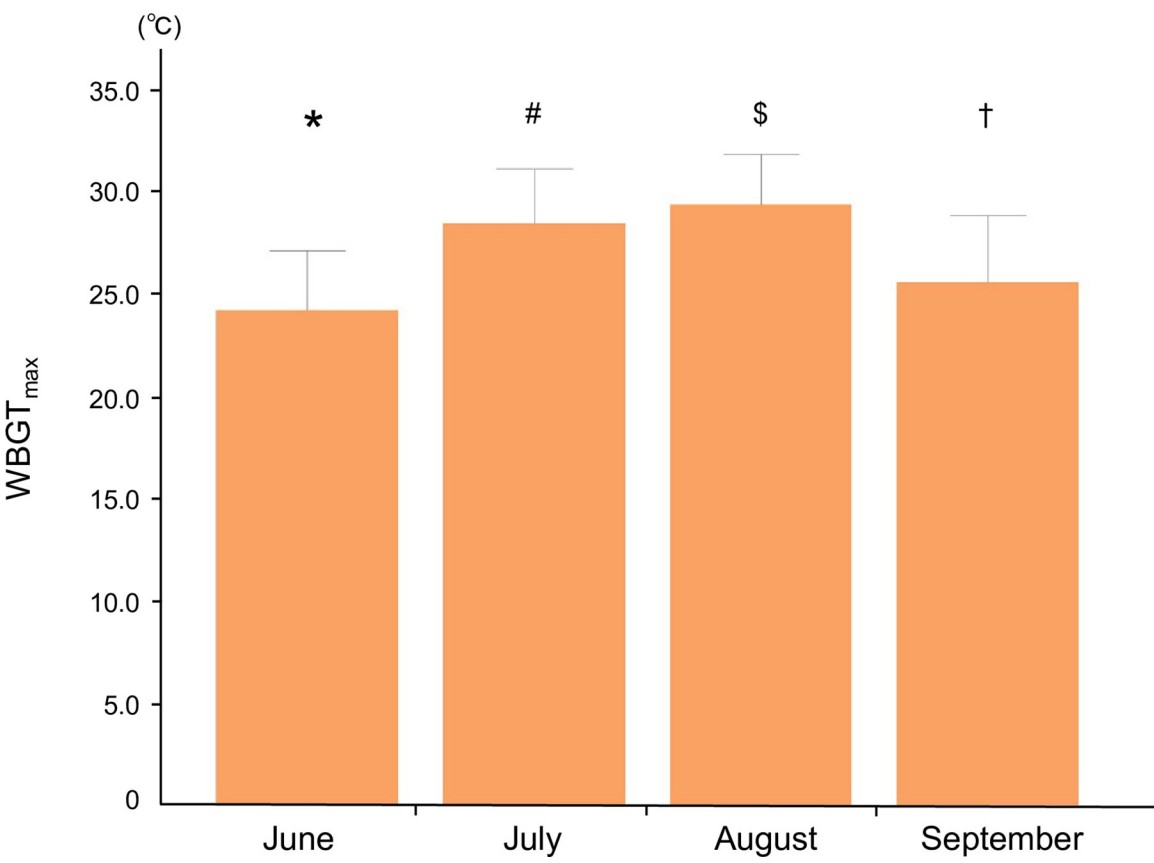

**Fig 4. The average WBGT$_{max}$ by month.** $*$ $p < 0.05$ vs July, August, and September. # $p < 0.05$ vs June, August, and September. $\$$ $p < 0.05$ vs June, July, and September. † $p < 0.05$ vs June, July, and August.

departments within each city. Another possible reason is that the examined area in the previous study had a lower percentage of older patients per 100,000 people compared to other areas with a higher percentage in Japan. Thus, since our study covered all regions of Japan, there was a great likelihood that the two studies would yield different percentages regarding older adults.

One of the new findings of this study is the significant difference by month in ADRHI_ALL. Specifically, the highest frequencies of ADRHI_ALL (Fig 6) and ADRHI_OLD were observed in August while the lowest frequencies were observed in June. Those of ADRHI_YA in July and August significantly differed from the other months, whereas those in June and September were not significantly different. The primary reason for these significant differences among the months may be related to meteorological condition variations throughout the year. Indeed, the average WBGT$_{max}$ in this study was 24.1 ± 2.9°C in June, 28.4 ± 2.7°C in July, 29.3 ± 2.5°C in August, and 25.5 ± 3.3°C in September (Fig 4). This result is partly supported by previous findings, which described a causal relationship between the meteorological condition and the number of heat illnesses [7, 8, 17, 18].

The current result also indicated the potential for the relationship between heat acclimation and ADRHI. Notably, the first steep increase in ADRHI_ALL in June was seen at a WBGT$_{max}$ of 23°C, while that in July and August were at a WBGT$_{max}$ of 26°C and 27°C, respectively. It is well known that factors other than weather are involved in the occurrence of heat illness, such as economic conditions, health conditions, and heat acclimation [3, 8]. Among these factors,

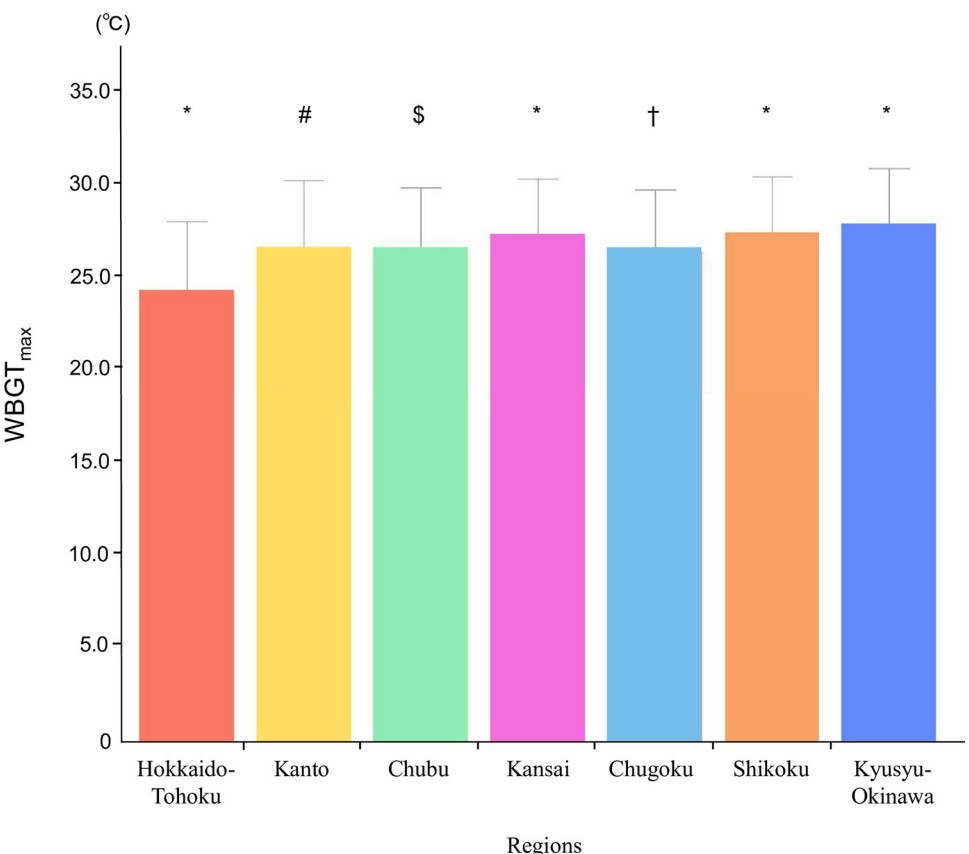

**Fig 5. The average WBGT$_{max}$ by region.** * $p < 0.05$ vs all other regions. # $p < 0.05$ vs all other regions except for Chubu. $ $p < 0.05$ vs all other regions except for Kanto and Chugoku. † $p < 0.05$ vs all other regions except for Chubu.

heat acclimation is the best-known protection against heat illnesses [19, 25]. In general, heat acclimation requires several days of heat exposure [25] or exposure and exercise. As heat acclimation helps to decrease heat stress for the body, it also supports the reduction of the physiological burden that is caused by such weather conditions. Conversely, previous studies suggested that older people are likely to have modest improvement following the heat acclimation program, such as sweat rate response [16] or body fluid regulation [26]. For these reasons, the difference in the first steep increase of WBGT may be assumed to be based on age categories; however, that of the WBGT in June was the same between ADRHI_OLD and ADRHI_YA at 23°C WBGT$_{max}$. These results suggested that, at least in June, the hazard levels in Japan should be considered based on not being sufficiently prepared to deal with heat stress because of insufficient heat exposure and heat acclimation, regardless of age. Although there were no data regarding heat acclimation in our study, it can be inferred that heat acclimation may be important for preventing heat illnesses, and shifting the hazard level might serve as an early precautionary measure when warm weather is forecasted in early summer.

While a significant difference was observed in the number of ADRHI among months (Fig 6), no significant differences were found across the regions. However, when analyzed across each region, the first steep increase of WBGT$_{max}$ was a maximum of 4°C difference between regions. For example, the first steep increase in ADRHI_ALL in Hokkaido-Tohoku (see Table 3) was at 24°C, while that of Kyushu-Okinawa was at 28°C. This trend was also observed in the other categories, that is, ADRHI_OLD and ADRHI_YA (Table 3). Additionally, the

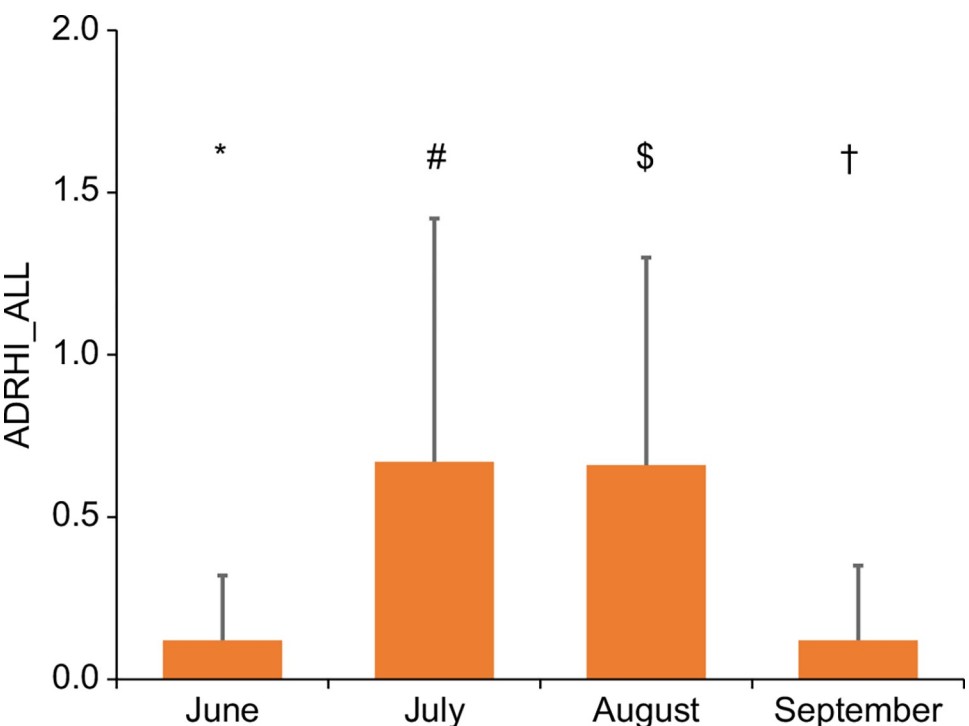

**Fig 6. Changes in ADRHI_ALL by month.** * p < 0.05 vs July, August, and September. # p < 0.05 vs June, August, and September. $ p < 0.05 vs June, July, and September. † p < 0.05 vs June, July, and August.

relative risks by Poisson regression analysis indicated that the values differed for each region (Table 4). This result was similar to a previous study that revealed different relative risks in the Kanto area in Japan [10]. In contrast, there was no significant difference in ARDHI across regions in this study, supporting a previous finding that the temperature at which heat-related mortality started to increase was lower at higher latitudes (i.e., north Finland), but the trend of the increase in daily mortality rate above that temperature was not significant between regions, such as in north Finland, London, and Athens [5]. The results of the previous study and the current study could suggest the mechanism of heat illness; that is, the fact that when the temperature rises heatstroke occurs. To support this result, when the ADRHI is summarized based on the Japanese heat hazard classification in each month for each region (Fig 8), about 64.1% of the ADRHI that occurred in June in the Hokkaido-Tohoku region was related to a $WBGT_{max}$ of $> 25°C$, which categorized "Caution" as the 2nd lowest level of the hazard rank, whereas only 15.0% of ARDHI occurred at the "Caution" level in the Kyushu-Okinawa region. Thus, these results indicated the need to adjust the hazard classification based on the actual occurrence of ARDHI in each region and each month.

Previous studies have suggested that such heat warning systems are important to trigger preventive actions to deal with heat illnesses [13, 27, 28]. Fouillet et al. [13] found that during a 2006 heat wave in France, and given the characteristics of the heat wave, there were approximately 4,400 fewer observed excess deaths than expected; the authors suggested that this finding may have been related to the implementation of adaptive measures, including an early heat warning system that was developed following the severe European heat wave of 2003. Meanwhile, Weinberger et al. [9] reported the effectiveness of NWS heat alerts in preventing mortality across 20 American cities, concluding that the NWS heat alerts were not associated with lower mortality in most cities because of the missed opportunity to implement a heat response

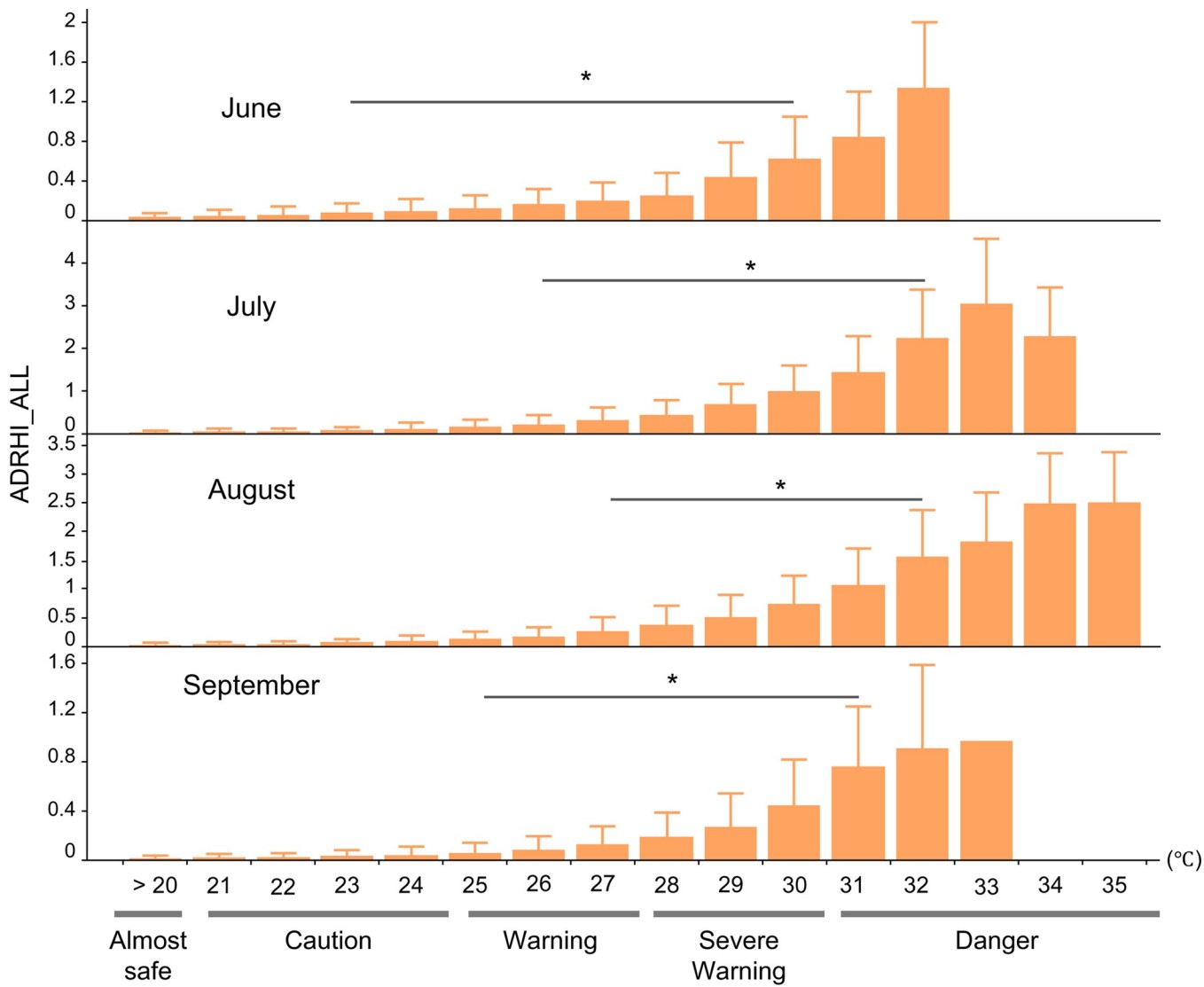

**Fig 7. Changes in ADRHI_ALL stratified by month and WBGT$_{max}$ from <21˚C to 35˚C.** *p < 0.05 vs previous WBGT$_{max}$.

plan to avert heat-related deaths. For example, the heat alert only covered 6% of the actual eligible days of the alert in Fayetteville, NC. In addition, other than preventing action for heat illness, previous studies also showed the significance of campaigns for stroke by the National

**Table 3. The first steep increase in ADRHI_ALL, ADRHI_OLD, and ADRHI_YA at WBGT$_{max}$ in each region.**

| Regions | ADRHI_ALL (˚C) | ADRHI_OLD (˚C) | ADRHI_YA (˚C) |
|---|---|---|---|
| Hokkaido-Tohoku | 24 | 24 | 25 |
| Kanto | 25 | 26 | 26 |
| Chubu | 26 | 26 | 27 |
| Kansai | 25 | 26 | 26 |
| Chugoku | 25 | 26 | 26 |
| Shikoku | 27 | 28 | 28 |
| Kyushu-Okinawa | 28 | 28 | 28 |

**Table 4. The relative risks for each region, month, WBGTmax, and proportion of people aged over 65.**

| Predictor | Relative risks |
|---|---|
| **Intercept** | **< 0.01** |
| **Regions** | |
| Ref. value Hokkaido-Tohoku | 1 |
| **Kanto** | **.859 (.848 - .868)** |
| **Chubu** | **.748 (.740 - .757)** |
| **Kansai** | **.829 (.820 - .839)** |
| **Chugoku** | **.638 (.629 - .647)** |
| **Shikoku** | **.361 (.355 - .367)** |
| **Kyushu-Okinawa** | **.472 (.467 - .478)** |
| **Month** | |
| Ref. value September | 1 |
| **June** | **1.898 (1.87–1.92)** |
| **July** | **2.373 (2.35–2.40)** |
| **August** | **1.748 (1.73–1.77)** |
| **WBGTmax** | **1.492 (1.490–1.495)** |
| **Proportion of people aged over 65** | **.992 (.991 - .993)** |

The values in parentheses are in the 95% probability range.

Stroke Foundation of Australia [29]. These preventive actions or implementation of campaigns might contribute to having a large impact not only on a decrease in the number of patients or dispatches but also on the awareness for heat illness or stroke. In Japan, to prevent an excessive increase in heat illnesses, the Ministry of the Environment provides an online forecast of a

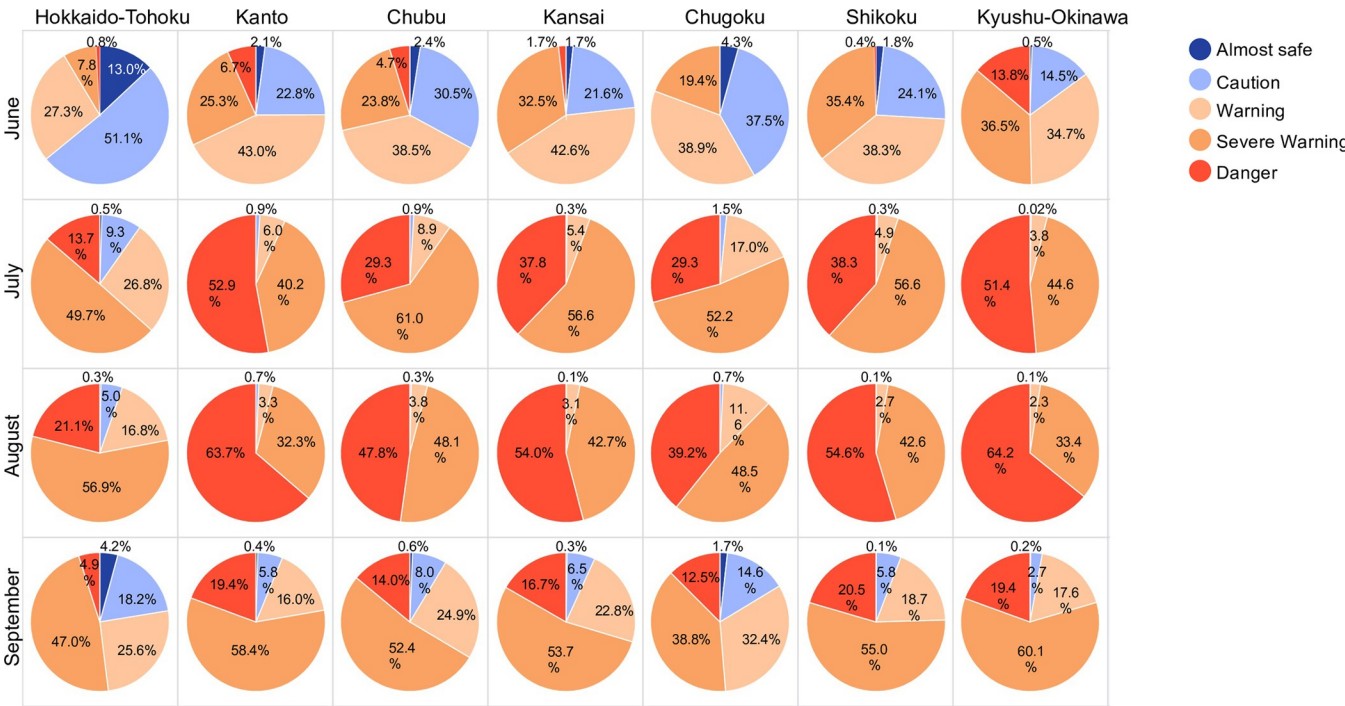

**Fig 8. ADRHI ratio in Japanese heat hazard classification by region and month.**

heat stress index based on the WBGT and information on the hazard levels of heat-related illnesses. However, the hazard classification of heat illnesses is based on a fixed set of criteria derived from the WBGT. In this guideline, a Warning occurs when the WBGT goes over < 25˚C and Danger when the WBGT goes over < 31˚C. This guideline, notwithstanding, may not capture the actual number of ADRHI which can occur at a lower temperature during the early period of warming weather, such as what happens in June in Japan. In fact, the ADRHI which occurred at a Danger classification in June accounted for 5.1% of all ADRHI in all regions; meanwhile, the ADRHI which occurred under a Caution classification (Almost safe and Caution) in the Hokkaido-Tohoku area in June accounted for 64.1% of all ADRHI in the region. Additionally, further analysis showed differences in the relative risks (Table 4), which implies the difference in risk when there is a one unit increase in WBGT$_{max}$. From these notions, to reduce the occurrence of heat illness, it is important: to set an optimal hazard or warning system that matches the actual occurrence of heat illness; to ensure that heat-related information is always up-to-date and accessible via, for example, the internet or a community information transportation tool; and to use this information for developing and implementing heat illness action plans, such as promoting the use of air conditioning, hydration, and measures to avoid an excessive increase in core temperatures [30] (e.g., to reduce heat illness). Nonetheless, in the month of June in Japan, it would likely be better for the hazard levels to be set to lower temperatures because the factors that influence the occurrence of heat illnesses are complex; such lowering of the threshold could make the heat warning system more effective. However, this study could not evaluate the actual effect of the hazard level on ARDHI; thus, future studies should investigate the awareness of the current hazard level and how it affects the implementation of action plans.

Our study has several limitations. First, our collected data did not allow for pinpointing the exact time and location of ADRHI occurrence; however, this does not mean that the occurrence of heat illnesses was unaffected by the local meteorological condition and did not differ by hours. Additionally, the WBGT data used were forecasted live using an estimated value calculated by the following formula; WBGT = 0.735×Ta+0.0374×RH+0.00292×Ta×RH+-7.619×SR−4.557×SR2−0.0572×WS−4.064 [31]. Moreover, we acknowledge that our analyzed data are not widely translatable because the occurrence of ADRHI depends on whether the individual is inside a house or not; particularly, an average of 39% of all ADRHI cases reportedly occurred inside the individual's house during 2017–2019 [12]. Second, we did not have access to factors other than WBGT data. As mentioned, several factors are assumed to be related to ADRHI, such as air pollution, economic status, and health status. However, we could not obtain data corresponding to heat stroke data for this experiment. Addressing this limitation is a concern for future research. Third, this study indicates that meteorological conditions other than WBGT, including daily maximum or apparent temperature, may increase ARDHI risk, particularly the time lag effect of maximum temperature or average apparent temperature [10]. However, this was because the current study aimed to investigate the relationship between the number of ARDHI and Japanese heat hazard levels based on WBGT. Despite several study limitations, our findings provided more realistic WBGT values, and it is believed that the data can help decision-makers in making more well-informed decisions about the development and implementation of action plans; this may ensure more effective measures against heat illnesses in Japan.

In summary, our results showed a significant relationship between ADRHI and WBGT$_{max}$ ($p < 0.05$, r = 0.74). Additionally, the number of ADRHI_ALL people showed significant differences by month, with the post hoc test detecting the first steep increase in ADRHI at a WBGT$_{max}$ of 23˚C when compared with that of 22˚C in June. For July, August, and September, the WBGT$_{max}$ were 26˚C, 27˚C, and 25˚C when compared with the previous WBGT$_{max}$

(Fig 7). In addition, the first significant increase in the number of ADRHI_ALL occurred at 24°C in Hokkaido-Tohoku, 25°C in Kanto, Kansai, and Chugoku, 26°C in Chubu, 27°C in Shikoku, and 28°C in Kyushu-Okinawa, compared with the previous WBGT$_{max}$. Further analysis revealed that the relative risks differed for each region and month. These results imply the need to update the hazard classification criteria, adjusting it by month and region.

## Acknowledgments

We thank Koya Suzuki (Faculty of Health and Sports Science, JUNTENDO University, Japan), Mr. Kazuhiro Tajima (NTT Com Online Marketing Solutions Corporation, Tokyo, Japan), and Satoshi Inose (Weathernews Inc, Chiba, Japan) for helpful statistical advice.

## Author Contributions

**Conceptualization:** Daisuke Nakamura, Kazuo Asada, Takayuki Ishiwata.

**Data curation:** Daisuke Nakamura, Hiroki Kinoshita.

**Formal analysis:** Daisuke Nakamura, Hiroki Kinoshita, Takuma Arimitsu, Takayuki Ishiwata.

**Investigation:** Daisuke Nakamura.

**Methodology:** Daisuke Nakamura, Takuma Arimitsu.

**Project administration:** Daisuke Nakamura.

**Supervision:** Mikinobu Yasumatsu, Takayuki Ishiwata.

**Visualization:** Daisuke Nakamura.

**Writing – original draft:** Daisuke Nakamura.

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
