## [Decision Letter · Decision Letter 0]

31 Aug 2021

PONE-D-21-20370

Trends in ambulance dispatches related to heat illness from 2010 to 2019; A secondary data analysis

PLOS ONE

Dear Dr. Nakamura,

Thank you for submitting your manuscript to PLOS ONE. After careful consideration, we feel that it has merit but does not fully meet PLOS ONE’s publication criteria as it currently stands. Therefore, we invite you to submit a revised version of the manuscript that addresses the points raised during the review process.

We look forward to receiving your revised manuscript.

Kind regards,

Ho Ting Wong, PhD

Academic Editor

PLOS ONE

Journal Requirements:

3. We note that you have included the phrase “data not shown” in your manuscript. Unfortunately, this does not meet our data sharing requirements. PLOS does not permit references to inaccessible data. We require that authors provide all relevant data within the paper, Supporting Information files, or in an acceptable, public repository. Please add a citation to support this phrase or upload the data that corresponds with these findings to a stable repository (such as Figshare or Dryad) and provide and URLs, DOIs, or accession numbers that may be used to access these data. Or, if the data are not a core part of the research being presented in your study, we ask that you remove the phrase that refers to these data

4. We note that Figure 1 in your submission contain map images which may be copyrighted. All PLOS content is published under the Creative Commons Attribution License (CC BY 4.0), which means that the manuscript, images, and Supporting Information files will be freely available online, and any third party is permitted to access, download, copy, distribute, and use these materials in any way, even commercially, with proper attribution. For these reasons, we cannot publish previously copyrighted maps or satellite images created using proprietary data, such as Google software (Google Maps, Street View, and Earth). For more information, see our copyright guidelines: http://journals.plos.org/plosone/s/licenses-and-copyright.

Reviewers' comments:

Reviewer's Responses to Questions

**Comments to the Author**

1. Is the manuscript technically sound, and do the data support the conclusions?

Reviewer #1: Partly

Reviewer #2: Yes

2. Has the statistical analysis been performed appropriately and rigorously? 

Reviewer #1: No

Reviewer #2: Yes

3. Have the authors made all data underlying the findings in their manuscript fully available?

Reviewer #1: No

Reviewer #2: Yes

4. Is the manuscript presented in an intelligible fashion and written in standard English?

Reviewer #1: Yes

Reviewer #2: Yes

5. Review Comments to the Author

Reviewer #1: The paper aimed to identify the association between the number of ambulance dispatches related to heat illness (ADRHI) and WBGT. And the authors want to propose the hazard classification needed to be adjusted in different regions. The idea is very good but the current results cannot support your conclusion so far. I would like to suggest some directions for authors to improve your results.

Major revision:

1. In your data, you have daily ADRHI data and also age group information. If you can apply zero-inflated Poisson model on your data, you can know how the effects on ADRHI by one unit increase of WBGT and also identify the effects from different age groups. This contribution will be very large because the heat illness might affect those vulnerable population. The alert signals can be issued to those specific population.

2. In addition, the regions have different weather conditions, you can stratify the models by seven regions and compare their effect’s differences. Then, you can propose different alerting model in different regions.

3. I will encourage the authors to use the daily data rather monthly data to show the effect from WBGT.

4. If you want to summarized the effects from seven regions, you can apply meta-regression to summarized their total effect.

Minor revision:

1. In the abstract, WBGT is needed to spell out the full name at the first occurrence.

2. The study design in the title can replace “a secondary data analysis” with “an ecological study”.

Reviewer #2: This study examines the relationship between ADHI and WBGT. I believe this is an important topic, with both academic and policy implications. I have the following suggestions to the authors:

1. Abstract: State the full name (wet bulb globe temperature) for WBGT in the abstract.

2. Methodology: Currently this analysis examines only two variables, ADHI and WBGT. But as the authors' have stated, there are many other factors that influence heat illness. The authors can consider using other statistical methods that can have multiple independent variables (including WBGT and others) and one dependent variable (ADHI), so that we can understand even when other factors are considered or controlled, WBGT still have positive relationship with ADHI.

3. other factors to be considered: this is related to point #2. I am wondering if the following factors also influence ADHI: (1) percentages of elderly, percentages of living alone, and/or percentages of elderly living alone. (2) the ratio between numbers of hospitals to population sizes (e.g., how many hosptials per, say, 1 million people). (3) poverty rate. (4) percentages of households/families that have air conditioners. (5) population density. (6) gender.

4. Heat awareness influence ADHI: The following paper suggest that heat awareness influence ADHI. Although the authors have listed this as limitations, I think that the authors can still think about how to incorporate this factor in their analysis.

https://www.ahajournals.org/doi/full/10.1161/STROKEAHA.110.612036

5. Lit review: Currently the manuscript does not have good discussion on the factors that influence heat illness or ADHI. I strongly suggest the authors to include this part, so that readers can evaluate the results of this study. For example, I am wondering whether the max daily temperature in one day or the duration of high temperature (high temp that last several days) matter more for heat illness. A proper lit review can clarify the questions.

6. fig 5: fig 5 suggests that the max temp do not result in most ADHI. That is very interesting, and the authors can elaborate more on this.

7.Fig 9: the analysis suggested that there are no regional differences between regions, but I am wondering how factors listed in point #3 influence the results. For example, how the percentage of elderly influence the results?

6. PLOS authors have the option to publish the peer review history of their article (what does this mean?). If published, this will include your full peer review and any attached files.

Reviewer #1: No

Reviewer #2: No

---

## [Author Response · Author response to Decision Letter 0]

24 Oct 2021

Reviewer #1

Trends in ambulance dispatches related to heat illness from 2010 to 2019: An ecological study

Thank you for your constructive review of our manuscript. Please note that our major revision of the statistical analysis is based on comments from both reviewers. The major revision is as follows:

1. We analyzed the ARDHI data in three groups—All population categories, older people categories (ADRHI_Old), and other than older people categories (ADRHI_YA).

2. To indicate the regional difference in ARDHI, we added the Poisson regression analysis.

3. We added the Fig.6.

The issues you pointed out were all very important. We have made revisions accordingly. Our point-by-point responses are written below each comment. We hope that these revisions will address your concerns.

We highlighted under the line and red font where we modified in the revised manuscript.

Reviewer #1 comments

The paper aimed to identify the association between the number of ambulance dispatches related to heat illness (ADRHI) and WBGT. And the authors want to propose the hazard classification needed to be adjusted in different regions. The idea is very good but the current results cannot support your conclusion so far. I would like to suggest some directions for authors to improve your results.

Major revision:

1. In your data, you have daily ADRHI data and also age group information. If you can apply zero-inflated Poisson model on your data, you can know how the effects on ADRHI by one unit increase of WBGT and also identify the effects from different age groups. This contribution will be very large because the heat illness might affect those vulnerable population. The alert signals can be issued to those specific population.

Ans.

Thank you for your very constructive comments. As you pointed out, we conducted Poisson regression analysis to investigate the effect on ARDHI by one degree of WBGT across each age category. The result of exponential β value is shown in Table 4 (P.16, L.255), and other parts for the explanation for the analysis. (P.10, L.183-185)

【P.10, L.183-185】

To analyze the relationship between ADRHI and WBGT, we performed Spearman’s correlation coefficient. To compare the differences for the number of ADRHI by month and region, we conducted Kruskal–Wallis test. When a significant difference was detected, to find where the first steep and significant increase in ADRHI occurred, a comparison circle was performed. Moreover, to investigate the effects on ADRHI by one unit increase of WBGT, Poison regression analysis was performed using SPSS (version 27.0; SPSS Inc, Chicago, IL). The statistical significance was set at p < 0.05. 

2. In addition, the regions have different weather conditions, you can stratify the models by seven regions and compare their effect’s differences. Then, you can propose different alerting model in different regions.

Ans.

Thank you for your pertinent comment. Similar to the answer above, we analyzed the data across each region and modified the paragraph in the discussion section related to the regional differences, from P.18 L.309 to P.19 L.338, to describe the importance of applying different alerting models by each region from this analysis. And we also added the Table.3 (P.15, L.248)

【P.18 L.309 to P.19 L.338】

These results suggested that, at least in June, the hazard levels in Japan should be considered based on not being sufficiently prepared to deal with heat stress because of insufficient heat exposure and heat acclimation. Although there were no data regarding heat acclimation in our study, it can be inferred that heat acclimation may be important for preventing heat illnesses, and shifting the hazard level might serve as an early precautionary measure when warm weather is forecasted in early summer.

While a significant difference was observed in the number of ADRHI for July and August compared to June and September (Fig. 7), no significant differences were found across the regions. However, when analyzed across each region, the first steep increase of WBGTmax was a maximum of 2℃ apart between regions. For example, the first steep increase of ADRHI_ALL in Hokkaido-Tohoku and Chugoku (see Table 3), while that of Kanto, Kansai, Shikoku, and Kyushu-Okinawa was 28℃. This trend was also observed in the other categories, that is ADRHI_Old and ADRHI_YA (Table. 3). Additionally, to investigate the effects on ADRHI by one unit increase of WBGT, the exponential β value was calculated, and the values differed across each age category in each region (Table. 4). This result corroborates with a previous study that revealed different relative risks in Kanto area in Japan [10]. In contrast, there is no significant difference in ARDHI across regions in the current study, supporting a previous finding that the temperature at which heat-related mortality started to increase was lower at higher latitudes (i.e., north Finland), but the trend of the increase in daily mortality rate above that temperature was not significant between the regions such as in north Finland, London, and Athens [5]. The results of the previous study and the current study could suggest the mechanism of heat illness, that is, the fact that when the temperature rises heatstroke occurs. To support this result, when the ADRHI is summarized based on the Japanese heat hazard classification in each month for each region (Fig.9), about 64.1% of the ADRHI that occurred in June in the Hokkaido-Tohoku region was related to a WBGTmax of > 25°C, which categorized “Caution” as 2nd lowest level of the hazard rank, whereas only 15.0% of ARDHI occurred in the “Caution” at the Kyushu-Okinawa. Thus, these results indicated the need to adjust the hazard classification based on the actual occurrence of ARDHI in each region and each month.

3. I will encourage the authors to use the daily data rather monthly data to show the effect from WBGT.

Ans.

Thank you for this pertinent comment. As you pointed out, the changes in daily meteorological conditions should have an impact on the occurrence of ARDHI. On the other hand, this paper is focusing on the analysis of the trend of occurrence of each hazard level from the point of view of the big data. We agree to the importance of daily-based analysis on ARDHI, but the purpose of the current study was not to investigate the occurrence of ARDHI from the daily aspect.

However, we agree that future research requires an analysis based on your suggestion.

4. If you want to summarized the effects from seven regions, you can apply meta-regression to summarized their total effect.

Ans.

Thank you for your thoughtful suggestion. We agree that the use of meta-analysis is useful if we are to summarize the effects from seven regions. However, our study focused on the trend in the occurrence of each hazard level using big data, and showing the real number of ARDHI by each region as the first priority. We have added the result of the Poisson regression model to clearly indicate the regional difference in the occurrence of ARDHI.

Minor revision:

1. In the abstract, WBGT is needed to spell out the full name at the first occurrence.

Ans.

We have revised it accordingly (P2, L38)

【P2, L38】

ADRHI and wet bulb globe temperature (WBGT) data

2. The study design in the title can replace “a secondary data analysis” with “an ecological study”.

Ans.

Thank you for your suggestion. We modified the title from “Trends in ambulance dispatches related to heat illness from 2010 to 2019; A secondary data analysis” to “Trends in ambulance dispatches related to heat illness from 2010 to 2019: An ecological study” (See the title)

【See the title】

Trends in ambulance dispatches related to heat illness from 2010 to 2019: An ecological study

Reviewer #2 

Trends in ambulance dispatches related to heat illness from 2010 to 2019: An ecological study

Thank you for your very constructive review of our manuscript. Please note that our major revision of the statistical analysis is based on comments from both reviewers, the major revision was as follows:

1. We analyzed the ARDHI data in three groups—All population categories, older people categories (ADRHI_Old), and other than older people categories (ADRHI_YA).

2. To indicate the regional difference in ARDHI, we added the Poisson regression analysis.

3. We added the Fig.6.

All issues you pointed out were all very important. We have made revisions accordingly. Our point-by-point responses are written below each comment. We hope that these revisions will address your concerns.

We highlighted under the line and red font where we modified in the revised manuscript.

Comments by Reviewer #2

This study examines the relationship between ADHI and WBGT. I believe this is an important topic, with both academic and policy implications. I have the following suggestions to the authors:

1. Abstract: State the full name (wet bulb globe temperature) for WBGT in the abstract.

Ans.

We have expanded the full name accordingly (P2, L38)

【P2, L38】

ADRHI and wet bulb globe temperature (WBGT) data

2. Methodology: Currently this analysis examines only two variables, ADHI and WBGT. But as the authors' have stated, there are many other factors that influence heat illness. The authors can consider using other statistical methods that can have multiple independent variables (including WBGT and others) and one dependent variable (ADHI), so that we can understand even when other factors are considered or controlled, WBGT still have positive relationship with ADHI.

3. other factors to be considered: this is related to point #2. I am wondering if the following factors also influence ADHI: (1) percentages of elderly, percentages of living alone, and/or percentages of elderly living alone. (2) the ratio between numbers of hospitals to population sizes (e.g., how many hosptials per, say, 1 million people). (3) poverty rate. (4) percentages of households/families that have air conditioners. (5) population density. (6) gender.

Ans. (For the answer Q2 and Q3)

Thank you for your important suggestion. As you pointed out, several factors are related to the incidence of ARDHI, including poverty rate, the ratio between numbers of hospitals to population sizes, or percentages of households/families that have air conditioners. Unfortunately, we did not find these data in each prefecture and every year. We could only find the population data and WBGT data, and these data can be linked to the ARDHI data, which were used in the current study. In addition, we were also interested in the gender difference of ARDHI, but the data source of ARDHI did not include the information on gender. On the other hand, we actually performed multiple regression analysis to investigate factors affecting the occurrence of ARDHI. We used WBGT, Month, and the percentage of elderly people as an independent variable. However, the result was not interesting; WBGT and Month highly affected the ARDHI. If we have data on living situation, living alone or not or use of air condition or not, we can expect an interesting result if we apply multiple regression analysis; but, we could not find these data. Similar to your concern, we acknowledge that providing significant alerting information for heat illness as well as effects of other factors on the occurrence of ARDHI other than meteorological conditions is very important.

4. Heat awareness influence ADHI: The following paper suggest that heat awareness influence ADHI. Although the authors have listed this as limitations, I think that the authors can still think about how to incorporate this factor in their analysis.

Ans.

Thank you for your useful comment and suggestion. We have deleted the description on the awareness of “the potential effect of implementing heat illness prevention plans” and added information based on your suggestion (P20, L350-355).

【P20, L350-355】

In addition, other than preventing action for heat illness, previous studies also showed the significance of campaigns for stroke by the National Stroke Foundation of Australia [29]. These preventive actions or campaigns implementation might contribute to having a large impact not only on a decrease in the number of patients or dispatches but also on the awareness for heat illness or stroke.

5. Lit review: Currently the manuscript does not have good discussion on the factors that influence heat illness or ADHI. I strongly suggest the authors to include this part, so that readers can evaluate the results of this study. For example, I am wondering whether the max daily temperature in one day or the duration of high temperature (high temp that last several days) matter more for heat illness. A proper lit review can clarify the questions.

Ans.

Thank you for your meaningful suggestion. As you mentioned above, the occurrence of heat illness related to the max daily temperature in one day or the duration of high temperature (high temp that lasts several days) matters more for heat illness, and this hypothesis is supported by several studies. Actually, in the current study, there is a significant relationship between ARDHI and -1day WBGTmax and similar results in the main text. However, because the purpose of the current study was to investigate the relationship between the number of ARDHI and Japanese heat hazard level using WBGT, we could not incorporate your suggestion; rather, we recommend it for future studies.

 (P.22 L395-404) 

【P.22 L395-404】

Third, the current study indicates that meteorological conditions other than WBGT, including daily maximum or apparent temperature, may increase ARDHI risk; particularly the time lag effect of maximum temperature or average apparent temperature [10]. However, this was because the current study aimed to investigate the relationship between the number of ARDHI and Japanese heat hazard levels based on WBGT. Despite several limitations of the study, our findings provided more realistic WBGT values, and it is believed that the data can help decision makers in making more well-informed decisions about the development and implementation of action plans; this may ensure more effective measures against heat illnesses in Japan.

6. fig 5: fig 5 suggests that the max temp do not result in most ADHI. That is very interesting, and the authors can elaborate more on this.

Ans.

Thank you for your comment, we elaborated on this in the main manuscript. (P.17 L.289 to P.18 L.294)

【P.17 L.289 to P.18 L.294】

Interestingly, despite the strong relationships between ARDHI and WBGT (Fig. 4), most ADRHI did not occur in the highest WBGTmax (Fig.6). This corresponds with each of the given WBGT. For example, a total of 2 days were < 35℃ WBGTmax during the study period while 28℃ WBGTmax was observed for 39739 days. Additionally, the ratio of ARDHI per day in each case was 182 (364 cases/2 days) and 8.0 (39739 cases/4958 days). 

7.Fig 9: the analysis suggested that there are no regional differences between regions, but I am wondering how factors listed in point #3 influence the results. For example, how the percentage of elderly influence the results?

Ans.

Thank you for your constructive comments. First, to clarify the age category effect, we analyzed the ARDHI data in three categories, ADRHI_ALL, ADRHI_Old, and ADRHI_YA.

In addition, as the other reviewer also suggested, we performed Poisson regression analysis to clarify the effect on ARDH by one unit increase of WBGT across each age category. As expected, the result of the Poisson regression analysis shows the difference of exponential β value across each region and age category. This result is shown in Table. 4. (P.16 L.255) 

However, the first steep increase in WBGTmax in ARDHI showed no significant difference between ADRHI_Old, and ADRHI_YA in June (P18, L.304-314). We added these very meaningful results in line with your pertinent comment. We appreciate your comments.

【P18, L.304-314】

Conversely, previous studies suggested that Older people are likely to have modest improvement following the heat acclimation program, such as sweat rate response [16] or body fluid regulation [26]. For these reasons, it is likely to the difference in first steep increase WBGT between the age categories, however, that of the WBGT in June was the same between ADRHI_Old and ADRHI_YA at 24℃ WBGTmax. These results suggested that, at least in June, the hazard levels in Japan should be considered based on not being sufficiently prepared to deal with heat stress because of insufficient heat exposure and heat acclimation. Although there were no data regarding heat acclimation in our study, it can be inferred that heat acclimation may be important for preventing heat illnesses, and shifting the hazard level might serve as an early precautionary measure when warm weather is forecasted in early summer.

---

## [Decision Letter · Decision Letter 1]

15 Nov 2021

PONE-D-21-20370R1Trends in ambulance dispatches related to heat illness from 2010 to 2019: An ecological studyPLOS ONE

Dear Dr. Nakamura,

Thank you for submitting your manuscript to PLOS ONE. After careful consideration, we feel that it has merit but does not fully meet PLOS ONE’s publication criteria as it currently stands. Therefore, we invite you to submit a revised version of the manuscript that addresses the points raised during the review process.

We look forward to receiving your revised manuscript.

Kind regards,

Ho Ting Wong, PhD

Academic Editor

PLOS ONE

Journal Requirements:

Additional Editor Comments (if provided):

After the first round of revision, a lot of problems can still be observed in the manuscript. Some are listed below.

1. Table 2: it is not necessary to show both Total ADRHI and ADRHI/year, as they are just related by a factor of 1/10.

2. Table 2: the name of the first row should be clearer, it is difficult to remember the difference between Total ADRHI and ADRH_ALL/ year. The variable names used in the statistical software should be translated into human-readable names for putting on the manuscript.

3. Figure 3, it is difficult to distinguish between the colors, perhaps bar charts will be better.

4. Figures 5 and 6, again, the figures are atypical graphs that are not easy to be understood by general readers.

5. Line 203-219: The paragraphs are not clear which have to be rewritten.

6. Figure 7, and Line 210: they are talking about two different things.

7. Table 4, 95%CI and significant levels should be given.

In general, the manuscript should be amended to reduce the graphs and tables which are not necessary, the current style is not smooth enough.

Reviewers' comments:

Reviewer's Responses to Questions

**Comments to the Author**

1. If the authors have adequately addressed your comments raised in a previous round of review and you feel that this manuscript is now acceptable for publication, you may indicate that here to bypass the “Comments to the Author” section, enter your conflict of interest statement in the “Confidential to Editor” section, and submit your "Accept" recommendation.

Reviewer #1: All comments have been addressed

Reviewer #2: All comments have been addressed

2. Is the manuscript technically sound, and do the data support the conclusions?

Reviewer #1: Yes

Reviewer #2: Yes

3. Has the statistical analysis been performed appropriately and rigorously? 

Reviewer #1: Yes

Reviewer #2: Yes

4. Have the authors made all data underlying the findings in their manuscript fully available?

Reviewer #1: Yes

Reviewer #2: Yes

5. Is the manuscript presented in an intelligible fashion and written in standard English?

Reviewer #1: Yes

Reviewer #2: Yes

6. Review Comments to the Author

Reviewer #1: The authors have fully addressed my concerns.

One minor suggestion:

1. The variables' name (i.e. ADRHI_ALL, ADRHI_Old, and ADRHI_YA) can be removed from the abstract.

Reviewer #2: (No Response)

7. PLOS authors have the option to publish the peer review history of their article (what does this mean?). If published, this will include your full peer review and any attached files.

Reviewer #1: No

Reviewer #2: No

---

## [Author Response · Author response to Decision Letter 1]

29 Nov 2021

Reviewer #1

Trends in ambulance dispatches related to heat illness from 2010 to 2019: An ecological study

Thank you again for your careful review of our manuscript. We modified the manuscript based on your comments as indicated below.

1. The variables' name (i.e. ADRHI_ALL, ADRHI_Old, and ADRHI_YA) can be removed from the abstract.

Response

Per our suggestion, we deleted the variables' names, i.e., ADRHI_ALL, ADRHI_Old, and ADRHI_YA from the abstract.

---

## [Decision Letter · Decision Letter 2]

7 Feb 2022

PONE-D-21-20370R2Trends in ambulance dispatches related to heat illness from 2010 to 2019: An ecological studyPLOS ONE

Dear Dr. Nakamura,

Thank you for submitting your manuscript to PLOS ONE. After careful consideration, we feel that it has merit but does not fully meet PLOS ONE’s publication criteria as it currently stands. Therefore, we invite you to submit a revised version of the manuscript that addresses the points raised during the review process.

We look forward to receiving your revised manuscript.

Kind regards,

Ho Ting Wong, PhD

Academic Editor

PLOS ONE

Journal Requirements:

Additional Editor Comments (if provided):

I have also made some minor comments in addition to reviewer 1's comments

Line 181: What is a comparison circle?

Figure 5: it is better to show all months. This is because you mentioned the difference between months in the main text, but the figure does not show the difference. The situation is the same in Figure 7.

Figure 6: You mentioned that some places have the difference, but the figure shows that all the places are quite similar. How about you highlight the difference in the figure and let readers know the statistics test result?

For figure 8, it is not good for readers to guess the meaning of “*” because you are showing people a figure instead of a typical statistical table.

Reviewers' comments:

Reviewer's Responses to Questions

**Comments to the Author**

1. If the authors have adequately addressed your comments raised in a previous round of review and you feel that this manuscript is now acceptable for publication, you may indicate that here to bypass the “Comments to the Author” section, enter your conflict of interest statement in the “Confidential to Editor” section, and submit your "Accept" recommendation.

Reviewer #1: All comments have been addressed

Reviewer #2: All comments have been addressed

2. Is the manuscript technically sound, and do the data support the conclusions?

Reviewer #1: Yes

Reviewer #2: Yes

3. Has the statistical analysis been performed appropriately and rigorously? 

Reviewer #1: Yes

Reviewer #2: Yes

4. Have the authors made all data underlying the findings in their manuscript fully available?

Reviewer #1: Yes

Reviewer #2: Yes

5. Is the manuscript presented in an intelligible fashion and written in standard English?

Reviewer #1: Yes

Reviewer #2: Yes

6. Review Comments to the Author

Reviewer #1: Minor suggestion:

Abstract

1.(Old), (YA) can be removed from the abstract.

2. β value can be replaced by "the coefficients".

In Table 4: Old means "elderly"? Please change the term and define the age of elderly in the table's footnote.

Reviewer #2: (No Response)

7. PLOS authors have the option to publish the peer review history of their article (what does this mean?). If published, this will include your full peer review and any attached files.

Reviewer #1: No

Reviewer #2: No

---

## [Author Response · Author response to Decision Letter 2]

7 Mar 2022

Reviewer #1

Trends in ambulance dispatches related to heat illness from 2010 to 2019: An ecological study

Thank you again for your careful review of our manuscript. We modified the manuscript based on your comments as indicated below.

1.　(Old), (YA) can be removed from the abstract.

Response

We have removed (Old), (YA) from the abstract.

【Abstract】

Moreover, several studies have reported an age difference in thermoregulation between older and younger adults, implying that the hazard classification should also consider age differences. 　　

2. β value can be replaced by "the coefficients".

Response

We have replaced β value with the coefficients in the manuscript. (L.49, 245, 253, 322, 367, 414.)

・Further, Poisson regression analysis revealed that the exponential of the coefficients differed（L.49）　

・Moreover, the exponential of the coefficients differed in each region and in each age category (Table. 4). （L.243）　

・Table. 4 The exponential of the coefficients by each region and age（L.251）　

・Additionally, to investigate the effects on ADRHI by one unit increase of WBGT, the exponential of the coefficients was calculated, （L.315）　

・　Additionally, further analysis showed differences in the exponential of the coefficients (Table. 4) （L.360）　

・Further analysis revealed that the exponential of the coefficients differed in each region and age category. （L.407）　

3.　In Table 4: Old means "elderly"? Please change the term and define the age of elderly in the table's footnote.

Response

In the text, we have defined “Old” as those aged 65 years old and above, along with the definition for all the age categories. Thus, we have used “Old” in all figures and tables. I hope this is sufficient.

---

## [Editor Report · Decision Letter 3]

24 Mar 2022

PONE-D-21-20370R3Trends in ambulance dispatches related to heat illness from 2010 to 2019: An ecological studyPLOS ONE

Dear Dr. Nakamura,

Thank you for submitting your manuscript to PLOS ONE. After careful consideration, we feel that it has merit but does not fully meet PLOS ONE’s publication criteria as it currently stands. Therefore, we invite you to submit a revised version of the manuscript that addresses the points raised during the review process.

Referring to Figure 6, you mentioned that * P<0.05 vs July, August, and September. However, visually the bar represents June looks the same as the bar represents September. I would like to confirm if there is any mistake. Moreover, please also have a check on the same issue on other figures.

Regarding Figure 7, please add the unit for the x-axis. (i.e. WBGTmax). Please also have a check on the same issue on other figures.

We look forward to receiving your revised manuscript.

Kind regards,

Ho Ting Wong, PhD

Academic Editor

PLOS ONE

Journal Requirements:

Additional Editor Comments:

Referring to Figure 6, you mentioned that * P<0.05 vs July, August, and September. However, visually the bar represents June looks the same as the bar represents September. I would like to confirm if there is any mistake. Moreover, please also have a check on the same issue on other figures.

Regarding Figure 7, please add the unit for the x-axis. (i.e. WBGTmax). Please also have a check on the same issue on other figures.
---

## [Editor Report · Decision Letter 4]

11 Apr 2022

PONE-D-21-20370R4Trends in ambulance dispatches related to heat illness from 2010 to 2019: An ecological studyPLOS ONE

Dear Dr. Nakamura,

Thank you for submitting your manuscript to PLOS ONE. After careful consideration, we feel that it has merit but does not fully meet PLOS ONE’s publication criteria as it currently stands. Therefore, we invite you to submit a revised version of the manuscript that addresses the points raised during the review process.

Regarding the problem in Figure 6, how about you explicitly write down the 95% CI in an appropriate location such as the main text or figure? I think this can make things transparent. Please do the same on other figures with the same problem.

We look forward to receiving your revised manuscript.

Kind regards,

Ho Ting Wong, PhD

Academic Editor

PLOS ONE

Journal Requirements:

Additional Editor Comments:

Regarding the problem in Figure 6, how about you explicitly write down the 95% CI in an appropriate location such as the main text or figure?

I think this can make things transparent.

Please do the same on other figures with the same problem.
---

## [Decision Letter · Decision Letter 5]

9 May 2022

PONE-D-21-20370R5Trends in ambulance dispatches related to heat illness from 2010 to 2019: An ecological studyPLOS ONE

Dear Dr. Nakamura,

Thank you for submitting your manuscript to PLOS ONE. After careful consideration, we feel that it has merit but does not fully meet PLOS ONE’s publication criteria as it currently stands. Therefore, we invite you to submit a revised version of the manuscript that addresses the points raised during the review process.

We look forward to receiving your revised manuscript.

Kind regards,

Ho Ting Wong, PhD

Academic Editor

PLOS ONE

**Additional Editor Comments:**

I am sorry that because the statistics part of your manuscript is still unclear to me, I have asked a statistical expert to future comment on your manuscript.

His comment is attached for your further action, please.

Thank you for your patience.

Reviewers' comments:

Reviewer's Responses to Questions

**Comments to the Author**

1. If the authors have adequately addressed your comments raised in a previous round of review and you feel that this manuscript is now acceptable for publication, you may indicate that here to bypass the “Comments to the Author” section, enter your conflict of interest statement in the “Confidential to Editor” section, and submit your "Accept" recommendation.

Reviewer #3: (No Response)

2. Is the manuscript technically sound, and do the data support the conclusions?

Reviewer #3: Yes

3. Has the statistical analysis been performed appropriately and rigorously? 

Reviewer #3: I Don't Know

4. Have the authors made all data underlying the findings in their manuscript fully available?

Reviewer #3: Yes

5. Is the manuscript presented in an intelligible fashion and written in standard English?

Reviewer #3: Yes

6. Review Comments to the Author

Reviewer #3: How was the “first significant increase” defined? Was it through statistical testing?

Line 136- June 2010 to September 2019 is 9 years and 122 days.

Table 1 is not directly relevant to the main result and can be a supplement table.

ADRHI_old and ADRHI_YA- how were they defined? Number of ADRHI in old or young per 100,000 old or young? That is, both the numerator and denominator are limited the subgroup (old or young)?

How was the Poison regression set up? Was ADRHI count used as the response variable? Has the model accounted for the difference in population size for different prefecture? Were separate models fitted for different regions and different age groups? What were the coefficients in Table 4? Coefficient for WBGT as a continuous variable? There be may be curvature in the relationship and WBGT-squared term may also be needed. Were any covariates included? Potential covariates of interest to include are region, the proportion of 65 and above in the population, some measurements of socioeconomic levels of the prefecture, year, month of the year. By fitting all the data in the same model you can perform statistical test to see if the effects of WBGT differ for different regions or differ for different month.

What happened in 2018? Are there explanations for the sudden increase of ADRHI? Is this due to a particular region?

Figure 1. maybe overlay the histogram with a line to indicate the average WBGT across the years to see if the changes correlate?

Figure 2. what is shown here? the proportion of ADRHI in each age category for each of the 10 years? This figure does not seem to contain much information and there was not much text in the manuscript describing the Figure either. Maybe it should be a supplement?

Figure 3 is very informative and it shows the relationship between ADRH and WBGT. Similar plots showing ADRH in older and younger populations will be helpful.

Figure 4 and 5 are not very informative to me. They only show the distribution of WBGT but the primary interest is the relationship between WBGT and ADRH. It will be more of interest to show ADRH vs WBGT stratified by month and ADRHI_ALL vs WBGT stratified by region instead.

Figure 7. How was the comparison done? Each ADHRI level vs the previous level? Is this how “first significant increase” was defined? Please add the details in the method section.

Figure 8 was mentioned in the discussion. The relevant text should be in the result section instead. In the Poisson regression model what if the heat hazard classification is used as a predictor, will there be an interaction between region and heat hazard classification?

7. PLOS authors have the option to publish the peer review history of their article (what does this mean?). If published, this will include your full peer review and any attached files.

Reviewer #3: No

---

## [Author Response · Author response to Decision Letter 5]

21 May 2022

Reviewer #3: 

Thank you for your review of our manuscript.

We have further modified our manuscript according to your suggestions.

We hope you are satisfied with our response.

To clearly indicate our revisions, the revised parts of the manuscript are in red colored font.

Q. 1．How was the “first significant increase” defined? Was it through statistical testing?

Response

Thank you for your comment. We have added a description of the definition of the first significant increase to the manuscript as follows:

Lines 175–177

To compare the differences for the number of ADRHI by month and region, we conducted Kruskal–Wallis test. When a significant difference was detected, the Dann-Bonferroni test was used for post hoc analysis. In addition, to clarify the relationship between ADRHI and WBGTmax, the WBGTmax of the first significant increase in ADRHI compared to the previous WBGTmax was defined as the first significant increase point. Moreover, to investigate the effects on ADRHI by one unit increase of WBGT, Poisson regression analysis was performed using SPSS (version 27.0; SPSS Inc, Chicago, IL). The statistical significance was set at p < 0.05.

Q. 2.

Line 136- June 2010 to September 2019 is 9 years and 122 days.

Response

We used data from the years 2010, 2011, 2012, 2013, 2014, 2015, 2016, 2017, 2018, and 2019. Hence, we considered the period to be 10 years.

Q. 3.

Table 1 is not directly relevant to the main result and can be a supplement table.

Response

As you pointed out, Table 1 is not directly related to the main result; however, this table is very important to understand the results of the study. Because the investigation of this research focused on the regional differences in ADRHI, we think Table 1 is needed to understand the geographical locations of the regions being analyzed. To clarify, each region’s location and the specific location data helps to understand the meteorological conditions, which are very important factors in the occurrence of heatstroke. Thus, we do not think this table should be a supplement. We hope you understand.

Q. 4. ADRHI_old and ADRHI_YA- how were they defined? Number of ADRHI in old or young per 100,000 old or young? That is, both the numerator and denominator are limited the subgroup (old or young)?

Response

Thank you for your comment. We have included definitions of ADRHI_Old and ADRHI_YA in the Methods section as follows:

Lines 168–170

The WBGT was divided per 1℃, from 20℃ to 35℃. The ADRHI was evaluated by absolute value and per 100,000 people (ADRHI_All), per 100,000 people aged 65 years and older (ADRHI_Old), and other than ADRHI_Old (ADRHI_YA).

Q.5

How was the Poison regression set up? Was ADRHI count used as the response variable? Has the model accounted for the difference in population size for different prefecture? Were separate models fitted for different regions and different age groups? What were the coefficients in Table 4? Coefficient for WBGT as a continuous variable? There be may be curvature in the relationship and WBGT-squared term may also be needed. Were any covariates included? Potential covariates of interest to include are region, the proportion of 65 and above in the population, some measurements of socioeconomic levels of the prefecture, year, month of the year. By fitting all the data in the same model you can perform statistical test to see if the effects of WBGT differ for different regions or differ for different month.

Response

Thank you for your comment. Before answering this question, we would like to provide an excerpt of the communication between another reviewer, the editor, and us.

Regarding the statistical analysis, we were provided some suggestions by Reviewer #2 during the first peer review process almost 6 months ago. We have attached our responses from that round of revisions below:

Q from Reviewer #2:

1. In your data, you have daily ADRHI data and also age group information. If you can apply zero-inflated Poisson model on your data, you can know how the effects on ADRHI by one unit increase of WBGT and also identify the effects from different age groups. This contribution will be very large because the heat illness might affect those vulnerable population. The alert signals can be issued to those specific population.

Response from us:

Thank you for your very constructive comments. As you pointed out, we conducted Poisson regression analysis to investigate the effect on ARDHI by one degree of WBGT across each age category. The result of exponential β value is shown in Table 4, and other parts for the explanation for the analysis. 

After the first revision, Reviewer #2 was satisfied with our response. The editor required clarifications in the 2nd review process 5 months ago as follows: 

Q from the Editor:

7. Table 4, 95%CI and significant levels should be given.

Response from us:

We added the 95%CI in Table 4 in parenthesis.

Regions All age categories Old Adults Adolescents Infants Newborns

Hokkaido-Tohoku 1.43

(1.43–1.43) 1.42

(1.42–1.43) 1.46

(1.45–1.47) 1.39

(1.37–1.40) 1.31

(1.27–1.35) NS

Kanto 1.56

(1.56–1.57) 1.56

(1.55–1.56) 1.61

(1.60–1.61) 1.5

(1.49–1.51) 1.39

(1.36–1.41) NS

Chubu 1.54

(1.53–1.54) 1.53

(1.53–1.54) 1.58

(1.57–1.59) 1.47

(1.46–1.49) 1.39

(1.35–1.43) NS

Kansai 1.56 

(1.55–1.56) 1.56

(1.55–1.57) 1.59

(1.58–1.60) 1.47

(1.46–1.49) 1.44

(1.40–1.48) NS

Chugoku 1.37

(1.37–1.38) 1.36

(1.35–1.37) 1.40

(1.39–1.41) 1.35

(1.34–1.37) 1.34

(1.28–1.40) No-ADRHI

Shikoku 1.55

(1.53–1.56) 1.53

(1.51–1.54) 1.60

(1.57–1.62) 1.50

(1.47–1.53) 1.37

(1.26–1.48) No-ADRHI

Kyushu-Okinawa 1.58

(1.57–1.58) 1.56

(1.55–1.57) 1.63

(1.62–1.64) 1.54

(1.52–1.56) 1.44

(1.38–1.50) NS

ALL 1.52

(1.52–1.52) 1.50

(1.57–1.57) 1.57

(1.57–1.57) 1.47

(1.46–1.47) 1.38

(1.37–1.40) 1.13

(1.02–1.26)

After the 2nd revision, Reviewer #2 had a few minor comments which were addressed 3 months ago as below:

Q from Reviewer #2:

2. β value can be replaced by "the coefficients".

Response from us:

We have replaced β value with the coefficients in the manuscript. 

・Further, Poisson regression analysis revealed that the exponential of the coefficients differed（Line 46）　

・Moreover, the exponential of the coefficients differed in each region and in each age category (Table 4). （Line 242）　

・Table 4 The exponential of the coefficients by each region and age（Line 253）　

・Additionally, to investigate the effects on ADRHI by one unit increase of WBGT, the exponential of the coefficients was calculated （Line 317）　

・　Additionally, further analysis showed differences in the exponential of the coefficients (Table 4) （Line 362）　

・Further analysis revealed that the exponential of the coefficients differed in each region and age category. （Line 410）　

Response

We hope that these communications have clarified your questions. Please find point-by-point responses to your comments below:

・Was ADRHI count used as the response variable?

Yes.

・Has the model accounted for the difference in population size for different prefecture?

Yes, we accounted for the population size in the prefectures; however, it was not realistic to collect data from all 47 prefectures. As mentioned above, this study focused on the regional differences in ADRHI; hence, we conducted analysis on each region. Regarding your comment, you proposed that Table 1 be moved to the supplement file; however, to include prefecture difference in the analysis, the information in Table 1 is important. Hence, we have retained Table 1 in the manuscript. We hope you understand our point of view.

・Were separate models fitted for different regions and different age groups? 

Table 4 presents our analysis of the different regions and age categories.

・What were the coefficients in Table 4?

Please refer to the communication with the other reviewer and editor presented above.

・Potential covariates of interest to include are region, the proportion of 65 and above in the population, some measurements of socioeconomic levels of the prefecture, year, month of the year.

Thank you for your comment. As mentioned above, this study aimed to investigate data using big data analysis across 10 years to reconsider the Japanese hazard classification for heatstroke in Japan. We acknowledge the importance of the potential covariates recommended by you to be included in the analysis. As you pointed out, the socioeconomic levels of the participants have the potential to affect ADRHI. However, we did not have these data. Similarly, we did not think an analysis by year was suitable for the purpose of this study, as we wanted to address the need to reconsider the Japanese hazard classification for heatstroke based on big data analysis, and not every year’s data.

Again, we thank you for your comments as an expert in statistical analysis. We hope you understand the standing purpose of this study and our point of view. We agree that further analysis, including the potential covariates you pointed out, are required. Accordingly, we have already added this to the limitation section of the Discussion as follows: 

(Lines 389–392)

As mentioned, several factors are assumed to be related to ADRHI, such as air pollution, economic status, and health status. However, we could not obtain heat stroke data for this experiment. Addressing this limitation is a concern for future research. 

Q.6

What happened in 2018? Are there explanations for the sudden increase of ADRHI? Is this due to a particular region?

Answer

Thank you for your comment. We are grateful for your careful review of our data. As you stated, ADRHI in 2018 was higher than in other years. One of the reasons for this was that the average WBGT was higher compared to other years. However, as you know, data were collected from June to September and ADRHI occurred for many reasons. Hence, the exact reason for the sudden increase of ADRHI in 2018 is not known. However, the weather conditions were related to ADRHI significantly; thus, we have added information to the main manuscript to clarify this point.

(Lines 268–269)

To the best of our knowledge, this was the first study to examine the relationship between ADRHI and WBGT in Japan using big data analysis. As expected, the number of ADRHI showed a significant relationship with WBGTmax (p < 0.05, r2 = 0.55; Fig 3). This result is similar to the previous study, which reported that r2 was 0.48 in three big prefectures in Japan [6]. During our experimental period, the average WBGTmax was highest in July 2018, while the number of ADRHI cases were highest in 2018. 

Q.7

Figure 1. maybe overlay the histogram with a line to indicate the average WBGT across the years to see if the changes correlate?

Response

Thank you for your suggestion. We have added a line denoting the average WBGT in Fig 1.

Q.8

Figure 2. what is shown here? the proportion of ADRHI in each age category for each of the 10 years? This figure does not seem to contain much information and there was not much text in the manuscript describing the Figure either. Maybe it should be a supplement?

Response

Thank you for your comment. As mentioned above, this study aimed to investigate the hazard classification in Japan, and we also wished to analyze the age differences in this context. Therefore, we believe that the proportion of the ADRHI in each age category is one of the main results in the study. It supports the importance of reconsidering the hazard classification based on the age categories. In addition, this study first demonstrated that the age proportion of ADRHI is similar across the years from the point of view of Japan as a whole. In previous studies that focused on heatstroke, the age categories are the most important component, because the ability of thermoregulation of the body is closely related to age. Hence, we think this figure is necessary in the main manuscript, rather than in the supplement.

Q.9

Figure 3 is very informative and it shows the relationship between ADRH and WBGT. Similar plots showing ADRH in older and younger populations will be helpful.

Response

Thank you for your suggestion. We were also interested in analyzing the relationships as per your suggestions. Hence, we conducted the analysis; however, the results were not significantly different between age groups. While we agree with your suggestion, we decided to exclude this information from the manuscript as it did not add much to our Discussion, while increasing the word count of our paper.

Q.10

Figure 4 and 5 are not very informative to me. They only show the distribution of WBGT but the primary interest is the relationship between WBGT and ADRH. It will be more of interest to show ADRH vs WBGT stratified by month and ADRHI_ALL vs WBGT stratified by region instead.

Response

Thank you for your comment.

In the field of research regarding heatstroke or mortality related to meteorological conditions, the data about the meteorological conditions is the most important to understand the occurrence of heatstroke. Hence, it is generally included in the Results because it is correlated with the incidence of heatstroke. [1-3]

We hope you understand our point of view and hope our response addresses your concerns. Regarding your latter suggestion, we analyzed the data and found that the ADRHI coefficient value was higher in July and August; however, this was related to the number of ADRHI and the range of WBGT compared to June or September. For further clarification, please refer to our response to Q9.

1. Armstrong BG, Chalabi Z, Fenn B, Hajat S, Kovats S, Milojevic A, et al. Association of mortality with high temperatures in a temperate climate: England and Wales. J Epidemiol Community Health. 2011;65: 340-345. doi: 10.1136/jech.2009.093161. PMID: 20439353.

2. Ng CF, Ueda K, Ono M, Nitta H, Takami A. Characterizing the effect of summer temperature on heatstroke-related emergency ambulance dispatches in the Kanto area of Japan. Int J Biometeorol. 2014;58: 941-948. doi: 10.1007/s00484-013-0677-4. PMID: 23700200. 

3. Basu R. High ambient temperature and mortality: a review of epidemiologic studies from 2001 to 2008. Environ Health. 2009; 8: 40. doi: 10.1186/1476-069X-8-40. PMID: 19758453.

Q.11

Figure 7. How was the comparison done? Each ADHRI level vs the previous level? Is this how “first significant increase” was defined? Please add the details in the method section.

Response

Thank you for your comment. Please refer to our response to Q1.

Q.12

Figure 8 was mentioned in the discussion. The relevant text should be in the result section instead. 

Response

Thank you for your comment. We have moved the text and Fig 8 to the Results section.

(Lines 245–249)

In June, approximately 64.1% of the ADRHI that occurred in the Hokkaido-Tohoku region were related to a WBGTmax of > 25°C, which is categorized as “Caution”—the 2nd lowest level in the hazard classification, whereas only 15.0% of ARDHI occurred in the “Caution” level at Kyushu-Okinawa. However, from July to September, most ADRHI occurred at levels over “Warning”.

Q.13

In the Poisson regression model what if the heat hazard classification is used as a predictor, will there be an interaction between region and heat hazard classification?

Response

Thank you for your comment. As mentioned in the main text, hazard classification was set regardless of region. On the other hand, the Poisson regression model was used as to clarify the differences between the regions. Thus, we did not check for interactions. As mentioned above, we examined the regional differences and monthly differences using big data in order to help with the reconsideration of Japanese hazard classification. Hence, we believe that investigating the interaction would not have served our purpose. I hope you understand our point of view.

---

## [Decision Letter · Decision Letter 6]

26 May 2022

PONE-D-21-20370R6Trends in ambulance dispatches related to heat illness from 2010 to 2019: An ecological studyPLOS ONE

Dear Dr. Nakamura,

Thank you for submitting your manuscript to PLOS ONE. After careful consideration, we feel that it has merit but does not fully meet PLOS ONE’s publication criteria as it currently stands. Therefore, we invite you to submit a revised version of the manuscript that addresses the points raised during the review process.

We look forward to receiving your revised manuscript.

Kind regards,

Ho Ting Wong, PhD

Academic Editor

PLOS ONE

Reviewers' comments:

Reviewer's Responses to Questions

**Comments to the Author**

1. If the authors have adequately addressed your comments raised in a previous round of review and you feel that this manuscript is now acceptable for publication, you may indicate that here to bypass the “Comments to the Author” section, enter your conflict of interest statement in the “Confidential to Editor” section, and submit your "Accept" recommendation.

Reviewer #3: (No Response)

2. Is the manuscript technically sound, and do the data support the conclusions?

Reviewer #3: (No Response)

3. Has the statistical analysis been performed appropriately and rigorously? 

Reviewer #3: (No Response)

4. Have the authors made all data underlying the findings in their manuscript fully available?

Reviewer #3: (No Response)

5. Is the manuscript presented in an intelligible fashion and written in standard English?

Reviewer #3: (No Response)

6. Review Comments to the Author

Reviewer #3: Some of my comments were address in this revision but not all. In particular I would like the following comments be addressed.

As the authors listed June 2010 to September 2019 is 10 years (or 10 122-day periods) not 10 years and 122 days.

Regarding the Poisson regression. All relevant information should show in the manuscript. I still have trouble figuring out how were the models set up. Was separate Poisson model used for each age category and each region combination? The cell for “ALL” and “All age categories” was the relative risk estimate for all data? Were any covariates adjusted for in the model?

The model with all the data included while adjusting for covariates such as age group, region, month, and potential interaction will be more informative and will help to clarify the relationship between WBGT and ARDHI. This is also consistent with the goal of this study. If there is an interaction between WBGT and month this supports the idea that the same WBGT will influence ARDHI differently for different month, hence the heat hazard classification needs to month specific. Similarly, the comment I made last time “In the Poisson regression model what if the heat hazard classification is used as a predictor, will there be an interaction between region and heat hazard classification?” will help us understand if the heat hazard classification needs to be region specific.

The authors mentioned that they “accounted for the population size in the prefect”. Please clarify how was this achieved in the model? It is conceivable that the number of ARDHI will be directly link to the population size. One way to account for the population size is to set the population size of each prefecture as an offset in the Poisson regression model.

7. PLOS authors have the option to publish the peer review history of their article (what does this mean?). If published, this will include your full peer review and any attached files.

Reviewer #3: No

---

## [Author Response · Author response to Decision Letter 6]

15 Jun 2022

Reviewer #3: 

Thank you for your review of our manuscript. We have gained valuable insights from your suggestions. As per your comments, we have performed Poisson analysis by setting the population size as the offset value; therefore, we used the absolute data for ADRHI in the analysis. Thus, the difference among regions is more clearly indicated, which is supported by another study result mentioned in the manuscript. We are deeply grateful for your expert knowledge on the Poisson regression analysis. 

However, regarding question 3, we are unable to adequately understand your concern. Therefore, kindly provide more explanation regarding the insufficiency of our analysis in meeting our study objective and the suggested model. 

We apologize for the inconvenience and deeply appreciate your assistance in this regard. 

The responses to your comments are presented below and the changes are marked in red font in the manuscript.

Q. 1．As the authors listed June 2010 to September 2019 is 10 years (or 10 122-day periods) not 10 years and 122 days.

Response

Thank you for your comment. We have modified the text as follows:

Consequently, the total data set comprised 57,340 ADRHI and contained dispatches from 47 prefectures for 10 years and 122-day period. (L.136)

Q. 2．

Regarding the Poisson regression. All relevant information should show in the manuscript. I still have trouble figuring out how were the models set up. Was separate Poisson model used for each age category and each region combination? The cell for “ALL” and “All age categories” was the relative risk estimate for all data? Were any covariates adjusted for in the model?

Response

Thank you for your comment. We apologize for the unclear description. We have modified the Methods section to clearly explain the statistical analysis process. (L177-181)

We have used the absolute value in the Poisson model, and the average population was set as the offset value. We hope that these changes address your concern.

Moreover, to investigate the effects on ADRHI by one unit increase of WBGT in each region, Poisson regression analysis was performed using SPSS (version 27.0; SPSS Inc, Chicago, IL). The ADRHI, WBGTmax, and the average population was set as an objective value, covariates, and the offset value, respectively. The statistical significance was set at p < 0.05.

Q. 3．

The model with all the data included while adjusting for covariates such as age group, region, month, and potential interaction will be more informative and will help to clarify the relationship between WBGT and ARDHI. This is also consistent with the goal of this study. If there is an interaction between WBGT and month this supports the idea that the same WBGT will influence ARDHI differently for different month, hence the heat hazard classification needs to month specific. Similarly, the comment I made last time “In the Poisson regression model what if the heat hazard classification is used as a predictor, will there be an interaction between region and heat hazard classification?” will help us understand if the heat hazard classification needs to be region specific.

Response

Thank you for your comment. Unfortunately, we are unable to clearly understand your suggestion. In Question 4, as per your suggestion, we have modified the analysis and the results seem to provide more information regarding the difference among regions for the occurrence of ARDHI. Moreover, we hope that these changes address some of your concerns raised here. Similar to your guidance in Question 4, we would appreciate it if you could suggest specific methods to help us address the issue mentioned here.

In particular, we do not understand the following sentence because the heat hazard classification was not set depending on each region, “will there be an interaction between region and heat hazard classification?” Could you explain this further and suggest methods of analysis to achieve this as you have done in Question 4: “One way to account for the population size is to set the population size of each prefecture as an offset in the Poisson regression model.” 

Q. 4．

 The authors mentioned that they “accounted for the population size in the prefect”. Please clarify how was this achieved in the model? It is conceivable that the number of ARDHI will be directly link to the population size. One way to account for the population size is to set the population size of each prefecture as an offset in the Poisson regression model.

Response

Thank you for your comment, we agree with your opinion that the number of ARDHI will be directly linked to the population size. Thus, in the Poisson model, we performed the analysis by setting the average population size as the offset value. (Table 4)

In addition, according to the modified analysis, we have revised the text describing the Poisson analysis.

L47 

across each region and age →　across each region

L245-L246

Moreover, the exponential of the coefficients differed in each region and in each age category (Table 4).　 

→　Moreover, the exponential of the coefficients differed for each region and in each age category (Table 4).

L328

the exponential of the coefficients was calculated, and the values differed across each age category in each region (Table 4).

　→　the exponential of the coefficients was calculated, and the values differed for each age category in each region (Table 4).　

L420

Further analysis revealed that the exponential of the coefficients differed in each region and age category.　

 →　Further analysis revealed that the exponential of the coefficients differed for each region and age category.

---

## [Decision Letter · Decision Letter 7]

27 Jun 2022

PONE-D-21-20370R7Trends in ambulance dispatches related to heat illness from 2010 to 2019: An ecological studyPLOS ONE

Dear Dr. Nakamura,

Thank you for submitting your manuscript to PLOS ONE. After careful consideration, we feel that it has merit but does not fully meet PLOS ONE’s publication criteria as it currently stands. Therefore, we invite you to submit a revised version of the manuscript that addresses the points raised during the review process.

We look forward to receiving your revised manuscript.

Kind regards,

Ho Ting Wong, PhD

Academic Editor

PLOS ONE

Journal Requirements:

Additional Editor Comments (if provided):

I am pleased to let you know that the statistical reviewer recommended a minor revision. As the decision of minor revision does not guarantee acceptance, it is also recommended that you should seek professional statistical support to help you revise the manuscript.

Reviewers' comments:

Reviewer's Responses to Questions

**Comments to the Author**

1. If the authors have adequately addressed your comments raised in a previous round of review and you feel that this manuscript is now acceptable for publication, you may indicate that here to bypass the “Comments to the Author” section, enter your conflict of interest statement in the “Confidential to Editor” section, and submit your "Accept" recommendation.

Reviewer #3: (No Response)

2. Is the manuscript technically sound, and do the data support the conclusions?

Reviewer #3: (No Response)

3. Has the statistical analysis been performed appropriately and rigorously? 

Reviewer #3: (No Response)

4. Have the authors made all data underlying the findings in their manuscript fully available?

Reviewer #3: (No Response)

5. Is the manuscript presented in an intelligible fashion and written in standard English?

Reviewer #3: (No Response)

6. Review Comments to the Author

Reviewer #3: I just realized that there was a typo in my original comment 1 for the last revision. I meant to say "9 years and 122 days" or "10 years" since from 2010 to 2019 (including both 2010 and 2019) are 10 years. Sorry about that.

Response to question 2:

"ADRHI, WBGTmax, and the average population was set as an objective value, covariates,

and the offset value, respectively."

"objective value" should be "response variable". "predictor" is probably a better choice than "covariates" here since WBGTmax is the only predictor in the model, right? Does "average population" mean "population size"?

Here is a little bit more details about Question 3- A multivariate Poisson regression can be used while fitting all data in the model. In addition to WBGTmax, the predictors can include other covariates such as age group, region, month and interaction terms between WBGTmax and age group, region, and month. A multiplicative interaction is basically a product term between two variables, e.g.,WBGTmax*age_group, WBGTmax*region, WBGTmax*month (note that age group, region and month should be categorical variables here). The interaction terms can tell you if the effect of WBGTmax differs across different categories of the interacting variables, e.g., effect of WBGTmax is different for different age group or not.

7. PLOS authors have the option to publish the peer review history of their article (what does this mean?). If published, this will include your full peer review and any attached files.

Reviewer #3: No

---

## [Author Response · Author response to Decision Letter 7]

2 Jul 2022

Reviewer #3: 

Thank you for your review of our manuscript and your kind advice and significant insight which has helped us improve the manuscript further. To respond to your request, we have consulted with a specialist in statistical analysis who majors in artificial intelligence and have now included him as one of the authors.

The responses to your comments are presented below and the changes are marked in red font in the manuscript.

Q1

I just realized that there was a typo in my original comment 1 for the last revision. I meant to say “9 years and 122 days” or “10 years” since from 2010 to 2019 (including both 2010 and 2019) are 10 years. Sorry about that.

Response

Thank you for your comment. We have modified the text as follows:

→

Consequently, the total data set comprised of 57,340 ADRHI and dispatches from 47 prefectures for 10 years. (L134)

Q2 

“ADRHI, WBGTmax, and the average population was set as an objective value, covariates,

and the offset value, respectively.”

“objective value” should be “response variable”. “predictor” is probably a better choice than “covariates” here since WBGTmax is the only predictor in the model, right? Does “average population” mean “population size”?

Response

Thank you for your comment. We have modified the text as follows:

→

The ADRHI, WBGTmax, and population size were set as the response value, predictor, and offset value, respectively. (L178)

Q3 

A multivariate Poisson regression can be used while fitting all data in the model. In addition to WBGTmax, the predictors can include other covariates such as age group, region, month and interaction terms between WBGTmax and age group, region, and month. A multiplicative interaction is basically a product term between two variables, e.g.,WBGTmax*age_group, WBGTmax*region, WBGTmax*month (note that age group, region and month should be categorical variables here). The interaction terms can tell you if the effect of WBGTmax differs across different categories of the interacting variables, e.g., effect of WBGTmax is different for different age group or not.

Response

Thank you for your detailed explanation of the analysis. According to your advice, we have obtained informative data to depict the regional differences. The value of Akaike's Information Criterion (AIC) and deviance was greatly decreased when a multivariate Poisson regression was conducted using WBGTmax and region, compared to when using WBGTmax and month. WBGTmax*region did not show good AIC and deviance. Hence, we added the comments on this in the discussion section. We hope this revision meets your request, and the right process and understanding of the data in multivariate Poisson regression has now been presented.

(L324-333)

→

Additionally, to investigate the effects on ADRHI by one unit increase of WBGT, the exponential of the coefficients was calculated, and the values differed for each region (Table 4). This result corroborates with a previous study that revealed different relative risks in Kanto area in Japan [10]. Moreover, the value of Akaike's Information Criterion (AIC) and deviance were greatly decreased when a multivariate Poisson regression was conducted using WBGTmax and region compared to that when using WBGTmax and month as predictors. Furthermore, it was indicated that the ratio of older people (> 65 years) was an important predictor of the effect of the value of AIC and deviance in multivariate Poisson regression. This result might lead to the consideration of hazardous levels, based on the age difference.

---

## [Decision Letter · Decision Letter 8]

19 Jul 2022

PONE-D-21-20370R8Trends in ambulance dispatches related to heat illness from 2010 to 2019: An ecological studyPLOS ONE

Dear Dr. Nakamura,

Thank you for submitting your manuscript to PLOS ONE. After careful consideration, we feel that it has merit but does not fully meet PLOS ONE’s publication criteria as it currently stands. Therefore, we invite you to submit a revised version of the manuscript that addresses the points raised during the review process.

We look forward to receiving your revised manuscript.

Kind regards,

Ho Ting Wong, PhD

Academic Editor

PLOS ONE

Journal Requirements:

Additional Editor Comments (if provided):

As the decision of minor revision does not guarantee acceptance, it is also recommended that you should seek professional statistical support to help you revise the manuscript.

Reviewers' comments:

Reviewer's Responses to Questions

**Comments to the Author**

1. If the authors have adequately addressed your comments raised in a previous round of review and you feel that this manuscript is now acceptable for publication, you may indicate that here to bypass the “Comments to the Author” section, enter your conflict of interest statement in the “Confidential to Editor” section, and submit your "Accept" recommendation.

Reviewer #3: (No Response)

2. Is the manuscript technically sound, and do the data support the conclusions?

Reviewer #3: (No Response)

3. Has the statistical analysis been performed appropriately and rigorously? 

Reviewer #3: (No Response)

4. Have the authors made all data underlying the findings in their manuscript fully available?

Reviewer #3: (No Response)

5. Is the manuscript presented in an intelligible fashion and written in standard English?

Reviewer #3: (No Response)

6. Review Comments to the Author

Reviewer #3: It is not clear how the Poisson regression model was set up. Have region, month, proportion of age over 65 been considered as covariates? They were mentioned in the discussion but not in the method or result. Have likelihood ratio tests been performed to see if each variable is statistically significant? What are the parameter estimates? The newly added paragraph in the discussion belongs to the result and more details about how the statistical modeling was performed need to be provided.

What is shown in Table 4? Poisson regression result? Was region the only predictor in the model? Risk estimates from the final Poisson model should be shown instead of a model without adjusting for other covariates.

7. PLOS authors have the option to publish the peer review history of their article (what does this mean?). If published, this will include your full peer review and any attached files.

Reviewer #3: No

---

## [Author Response · Author response to Decision Letter 8]

1 Aug 2022

Reviewer #3: 

Thank you for your thoughtful review of our manuscript. The responses to your comments are presented below and the changes are marked in red font in the manuscript.

Q1

 It is not clear how the Poisson regression model was set up. Have region, month, proportion of age over 65 been considered as covariates? They were mentioned in the discussion but not in the method or result. Have likelihood ratio tests been performed to see if each variable is statistically significant? What are the parameter estimates? The newly added paragraph in the discussion belongs to the result and more details about how the statistical modeling was performed need to be provided.

What is shown in Table 4? Poisson regression result? Was region the only predictor in the model? Risk estimates from the final Poisson model should be shown instead of a model without adjusting for other covariates.

Response

Thank you for your helpful comments and suggestions. We apologize that we did not understand your question correctly. We modified the text regarding the Poisson regression analysis in the method and result sections, and revised Table 4. We hope that this modification adequately addresses your concerns.

Method section (p. 176-178)

Moreover, to investigate the effects on ADRHI of a one unit increase in WBGT in each region, a Poisson regression analysis was performed using SPSS (version 27.0; SPSS Inc, Chicago, IL) to investigate the effects on ADRHI of a one unit increase in WBGT in each region. In the Poisson model, the ADRHI was set as the response value, and regions, month, WBGTmax, and proportion of people aged over 65 were set as a predictor. Population size was set as an offset value. The statistical significance was set at p < 0.05.

Results section (p. 239-241)

Moreover, the exponential values of the coefficients differed for each region and month are shown in Table 4 (LRχ2 = 697742.8, p < 0.001). The value of the exponential of the coefficients also differed among regions and months. 

Table 4.

Predictor b SE Waldχ2 The exponential of the coefficients

Intercept -9.63 .03 145630.9*** < 0.01

Regions 33902.3*** 

Ref. value

Hokkaido-Tohoku 0 1

Kanto -0.15 .01 685.7*** .859

Chubu -0.29 0.01 2536.2*** .748

Kansai -0.19 0.01 1066.5*** .829

Chugoku -0.45 0.01 4273.4*** .638

Shikoku -1.02 0.01 14950.8*** .361

Kyushu-Okinawa -0.75 0.01 14777.1*** .472

Month 29617.7*** 

Ref. value

September 0 1

June 0.64 0.01 7958.0*** 1.898

July 0.86 0.01 24603.3*** 2.373

August 0.56 0.01 9878.0*** 1.748

WBGTmax 0.40 0.00 280255.3*** 1.492

Proportion of people aged over 65 -0.01 0.00 330.2*** .992

***; P < .001.

---

## [Decision Letter · Decision Letter 9]

25 Aug 2022

PONE-D-21-20370R9Trends in ambulance dispatches related to heat illness from 2010 to 2019: An ecological studyPLOS ONE

Dear Dr. Nakamura,

Thank you for submitting your manuscript to PLOS ONE. After careful consideration, we feel that it has merit but does not fully meet PLOS ONE’s publication criteria as it currently stands. Therefore, we invite you to submit a revised version of the manuscript that addresses the points raised during the review process.

We look forward to receiving your revised manuscript.

Kind regards,

Ho Ting Wong, PhD

Academic Editor

PLOS ONE

Journal Requirements:

Reviewers' comments:

Reviewer's Responses to Questions

**Comments to the Author**

1. If the authors have adequately addressed your comments raised in a previous round of review and you feel that this manuscript is now acceptable for publication, you may indicate that here to bypass the “Comments to the Author” section, enter your conflict of interest statement in the “Confidential to Editor” section, and submit your "Accept" recommendation.

Reviewer #3: (No Response)

2. Is the manuscript technically sound, and do the data support the conclusions?

Reviewer #3: (No Response)

3. Has the statistical analysis been performed appropriately and rigorously? 

Reviewer #3: (No Response)

4. Have the authors made all data underlying the findings in their manuscript fully available?

Reviewer #3: (No Response)

5. Is the manuscript presented in an intelligible fashion and written in standard English?

Reviewer #3: (No Response)

6. Review Comments to the Author

Reviewer #3: In a Poisson regression the exponential of the coefficient is the relative risk. Table 4- the current columns are not very helpful for a paper. Maybe only show the point estimate of relative risk and its 95% confidence interval is enough. Given the large sample size, p values do not carry as much information as the 95% confidence intervals.

7. PLOS authors have the option to publish the peer review history of their article (what does this mean?). If published, this will include your full peer review and any attached files.

Reviewer #3: No

---

## [Author Response · Author response to Decision Letter 9]

25 Aug 2022

Reviewer #3: 

Thank you for your thoughtful review of our manuscript. The responses to your comments are presented below and the changes are marked in red font in the manuscript.

Q1

In a Poisson regression the exponential of the coefficient is the relative risk. Table 4- the current columns are not very helpful for a paper. Maybe only show the point estimate of relative risk and its 95% confidence interval is enough. Given the large sample size, p values do not carry as much information as the 95% confidence intervals.

Response

Thank you for your kind advice. We modified Table 4. (L252-266)

---

## [Decision Letter · Decision Letter 10]

12 Sep 2022

PONE-D-21-20370R10Trends in ambulance dispatches related to heat illness from 2010 to 2019: An ecological studyPLOS ONE

Dear Dr. Nakamura,

Thank you for submitting your manuscript to PLOS ONE. After careful consideration, we feel that it has merit but does not fully meet PLOS ONE’s publication criteria as it currently stands. Therefore, we invite you to submit a revised version of the manuscript that addresses the points raised during the review process.

We look forward to receiving your revised manuscript.

Kind regards,

Ho Ting Wong, PhD

Academic Editor

PLOS ONE

Journal Requirements:

Additional Editor Comments (if provided):

The reviewer request you make the following minor change. Once you have amended your manuscript, I will make an acceptance decision. Thank you.

Please use "relative risk" instead of "exponential of coefficient" for the interpretation of the Poisson model throughout the paper.

Reviewers' comments:

Reviewer's Responses to Questions

**Comments to the Author**

1. If the authors have adequately addressed your comments raised in a previous round of review and you feel that this manuscript is now acceptable for publication, you may indicate that here to bypass the “Comments to the Author” section, enter your conflict of interest statement in the “Confidential to Editor” section, and submit your "Accept" recommendation.

Reviewer #3: All comments have been addressed

2. Is the manuscript technically sound, and do the data support the conclusions?

Reviewer #3: (No Response)

3. Has the statistical analysis been performed appropriately and rigorously? 

Reviewer #3: (No Response)

4. Have the authors made all data underlying the findings in their manuscript fully available?

Reviewer #3: (No Response)

5. Is the manuscript presented in an intelligible fashion and written in standard English?

Reviewer #3: (No Response)

6. Review Comments to the Author

Reviewer #3: Please use "relative risk" instead of "exponential of coefficient" for the interpretation of the Poisson model throughout the paper.

7. PLOS authors have the option to publish the peer review history of their article (what does this mean?). If published, this will include your full peer review and any attached files.

Reviewer #3: No

---

## [Author Response · Author response to Decision Letter 10]

14 Sep 2022

Reviewer #3: 

Thank you for your thoughtful review of our manuscript. The responses to your comments are presented below and the changes are marked in red font in the manuscript.

Q1

Please use "relative risk" instead of "exponential of coefficient" for the interpretation of the Poisson model throughout the paper.

Response

Thank you for your kind advice. We used "relative risk" instead of "exponential of coefficient" for the interpretation of the Poisson model throughout the paper.

---

## [Editor Report · Decision Letter 11]

21 Sep 2022

Trends in ambulance dispatches related to heat illness from 2010 to 2019: An ecological study

PONE-D-21-20370R11

Dear Dr. Nakamura,

We’re pleased to inform you that your manuscript has been judged scientifically suitable for publication and will be formally accepted for publication once it meets all outstanding technical requirements.

Kind regards,

Ho Ting Wong, PhD

Academic Editor

PLOS ONE
---

## [Editor Report · Acceptance letter]

28 Oct 2022

PONE-D-21-20370R11 

Trends in ambulance dispatches related to heat illness from 2010 to 2019: An ecological study 

Dear Dr. Nakamura:

I'm pleased to inform you that your manuscript has been deemed suitable for publication in PLOS ONE. Congratulations! Your manuscript is now with our production department. 

Kind regards, 

on behalf of

Dr. Ho Ting Wong 

Academic Editor

PLOS ONE